# Information Design in Multi-Agent Reinforcement Learning

**Yue Lin, Wenhao Li, Hongyuan Zha, Baoxiang Wang**[*]
The Chinese University of Hong Kong, Shenzhen
`linyue3h1@gmail.com, {liwenhao, zhahy, bxiangwang}@cuhk.edu.cn`

## Abstract

Reinforcement learning (RL) is inspired by the way human infants and animals learn from the environment. The setting is somewhat idealized because, in actual tasks, other agents in the environment have their own goals and behave adaptively to the ego agent. To thrive in those environments, the agent needs to influence other agents so their actions become more helpful and less harmful. Research in computational economics distills two ways to influence others directly: by providing tangible goods (*mechanism design*) and by providing information (*information design*). This work investigates information design problems for a group of RL agents. The main challenges are two-fold. One is the information provided will immediately affect the transition of the agent trajectories, which introduces additional non-stationarity. The other is the information can be ignored, so the sender must provide information that the receiver is willing to respect. We formulate the *Markov signaling game*, and develop the notions of signaling gradient and the extended obedience constraints that address these challenges. Our algorithm is efficient on various mixed-motive tasks and provides further insights into computational economics. Our code is publicly available at `https://github.com/YueLin301/InformationDesignMARL`.

## 1 Introduction

Reinforcement learning (RL) studies how a world-agnostic agent makes sequential decisions to maximize utility. It gains increasing popularity and achieves milestone progress in Atari [46, 22], Go [48], Poker [8, 9, 39], video games [2, 6], and bioinformatics [26], economics [59], etc. The canonical RL formulation requires the environment to be stationary, meaning that other agents could not respond to the ego agent's policy by adapting their own policies [50]. The assumption does not hold in real applications. Instead, the ego agent, in general, cannot dictate other agents and needs to influence others so their adaptation becomes more helpful and less harmful.

A substantial subarea of RL, multi-agent reinforcement learning (MARL), investigates the interaction and influence among multiple RL agents when they are placed in a shared environment. It has obtained promising results thanks to the generality and diversity of its settings. Examples are pure cooperative games and pure competitive games. The pure cooperative setting studies agents that work on a consentaneous goal. In this case, influencing other agents is relatively straightforward. The pure competitive setting studies zero-sum games. In this case, influencing other agents is less likely to be practical or even impossible for two-player games. In between these two extreme cases arises the large, more realistic but less charted area of mixed-motive MARL, where influencing other learning agents becomes a main challenge [31, 37, 30].

---

[*]Corresponding to: Baoxiang Wang

37th Conference on Neural Information Processing Systems (NeurIPS 2023).

Studies in computational economics have distilled different ways of directly influencing a rational, self-interested agent into two types: by providing tangible goods (*mechanism design*) and by providing information (*information design*) [52, 15]. For the former type, tangible goods align perfectly with RL rewards because rewards are natural and utilitarian. Several works in RL design algorithms for the ego agent to use the reward to incentivize other agents [57, 58, 59, 29]. For the latter type, the ego agent must possess information that is helpful to the other agent while not observed by the other agent. The ego agent then sends a message that partially reveals this information in the hope that the other agents respect the message and act in a way that benefits the ego agent. The key observation in information design is that the message needs to be respected, which indicates that the message, apart from benefiting the ego agent, must also benefit the receiving agent.

Two difficulties persist in modeling and solving information design with reinforcement learning. The first challenge is the issue of non-stationarity. The receiver's environment changes as the sender's signaling scheme is updated. If the sender uses policy gradient, it does not consider how modifications to its scheme affect the receiver's learning [17]. On top of this, the signal impacts not only the updating phase but also the sampling phase. In fact, the receiver uses the signals from the sender as part of its input. The signaling scheme will, therefore, directly affect the trajectory generation. This means that specific techniques in mechanism design, such as hyper-gradient, are unsuitable, and new methods need to be formulated [57].

The second challenge lies in how the message can potentially be respected by the receiver. The most general signal space is the set of all state subsets and is exponentially large. An analysis analogous to the *revelation principle* proves that there is an optimal signaling scheme that uses a signal space of the same size as the action space of the receiver, which leads to the classic *obedience constraints* and the linear program formulation of information design [28, 40]. However, under the revelation principle, a signal suggests an exact action, which means the receiver will be dictated by the sender if it respects the message. When both parties are reinforcement learning agents, such want of dictatorship does not build trust and respect between them and instead inevitably drives them to the equilibrium where all signals are ignored. Counter-intuitively, we find that the revelation principle can be removed under the learning context, and one will need to resort to more persuasive signaling schemes and the corresponding update methods.

This paper investigates how an informative ego sender could learn to influence a self-interested receiver with its informational advantage. We propose *Markov signaling game* for mixed-motive communication, where in each timestep, the sender encodes and sends a message to the receiver. After the signaling step, the receiver will act based on the message and its observation. We derive the *signaling gradient* to learn the signaling scheme that addresses the non-stationarity problem. This gradient considers the additional gradient chain from the receiver's policy and is proved to be an unbiased gradient estimation. It agrees with our intuition that the influence of the sender on the behavior of the receiver should be reflected in the gradient term. Based on the signaling gradient, we design the *extended obedience constraints* for incentive compatibility of the signaling scheme.[2] We further provide an approximation of the gradient of such constraints, which solves the second challenge because it is suitable for learning algorithms while preserving the optimum of the ego agent. Information design in MARL is then end-to-end differentiable and learnable with our algorithm.

Numerical experiments on `Recommendation Letter` and `Reaching Goals` demonstrate the efficacy of our approach. Extended discussions are provided for the method and the empirical results.

## 2 Preliminaries and Related Works

**Information Design** For information design, the core insight is to send messages to change the posterior beliefs of the receiver, which persuades it to take actions that benefit the sender [52, 15]. The canonical formulation considers a task that a sender wants to persuade a myopic receiver for one step [28]. The sender and the receiver share a prior distribution $P(s)$ over the state $s$, which affects both the payoffs of the sender $r^i(s, a)$ and the receiver $r^j(s, a)$ (some recent work lifts this assumption [60]). The sender needs to honestly commit its signaling scheme, which is a policy that determines the distribution of signals, to the receiver before the interaction. This is referred to as the commitment assumption.

---

[2]A mechanism is said to be incentive compatible if every participant's optimal strategy, given the strategies of others and the mechanism's rules, leads to an outcome that is desired by the mechanism designer.

The flow of the persuasion process is as follows: (1) The sender commits a signaling scheme to the receiver; (2) The environment generates a state $s$. The sender observes the state $s$ and then samples a message according to the distribution of the committed signaling scheme; and (3) The receiver receives the message, and then calculates a posterior and chooses an optimal action for itself. Given the current state and the receiver's chosen action, the sender and the receiver get rewards from the environment.

Based on an analysis similar to the revelation principle [28, 19], there is an optimal signaling scheme that does not require more signals than the number of actions available to the receiver. Thus it is without loss of generality for the sender to recommend an action directly to the receiver rather than sending a message. From the self-interested receiver's perspective, as long as it believes that the recommended actions are optimal in its posterior belief, it will follow the sender's advice. This kind of constraints of the sender's signaling scheme is called obedience constraints. When such constraints are satisfied, the signaling scheme will be incentive compatible, meaning that the receiver will follow the sender's advice. In this way, the process of information design can be modeled as a constrained optimization problem

$$\max_{\varphi} \mathbb{E}_{\varphi}[\, w^i(s,a)\,], \ \ \text{s.t.} \sum_s P(s) \cdot \varphi(a \mid s) \cdot [\, w^j(s,a) - w^j(s,a')\,] \geq 0, \forall a, a', \quad (1)$$

where $w^i(s,a)$ and $w^j(s,a)$ are the utility functions of the sender and the receiver respectively, $\varphi(a \mid s)$ is the sender's signaling scheme, and the problem is formulated as a linear program.

Information design is applicable to a vast array of real-world scenarios, including voter coalition formation [1], law enforcement deployment [21], price discrimination [4], etc. See [27] for a summary.

**Learning to Communicate** Communication learning is a significant subarea of MARL. Existing research primarily focus on fully cooperative settings [16, 49, 42]. Among the proposed methods in MARL with communication, DIAL [16] is the closest work to ours. It highlights the importance of the receiver's feedback to the sender, where the receiver updates its critic network and passes the gradient back to the sender's network. This also implies that the sender is assisting the receiver in evaluating the environment. The altruistic design of the sender is appropriate in fully cooperative scenarios, but might not extend to mixed-motive scenarios.

MARL communication under mixed-motive settings was attempted in [25]. The work uses social influence, which is defined via the Kullback-Leibler divergence of the receiver's policy update, to measure how persuasive is the sender's message. It is a pioneer attempt and demonstrates efficacy in several contexts, but the authors observed that "The listener agent is not compelled to listen to any given speaker. Listeners selectively listen to a speaker only when it is beneficial, and influence cannot occur all the time." This describes a lack of the obedience constraints, and motivates us to propose algorithms that are more effective and more general.

Related works on mechanism design, sequential information design, and emergent communication are deferred to Appendix A.

## 3 Markov Signaling Games

Consider a signaling game involving a sender and a receiver. The sender $i$ is assumed to have access to the global state $s \in S$, while the receiver $j$ makes decisions based only on its local observation $o \in O$ and received message $\sigma \in \Sigma$ from $i$. At each timestep $t$, the observation $o_t \in O$ is sampled by the emission function $q : S \to O$, where the information in $o_t$ is a proper subset of the information in $s_t$. We overload the notation and write $o_t \subset s_t$. The receiver's observation at each timestep is common knowledge between the sender and receiver. The sender's *informational advantage* over the receiver at each timestep $t$ is reflected by $s_t - o_t$. By the sender's informational advantage of the problem setting, assume that $\{s_t - o_t\}_{t \geq 0}$ affects $j$'s payoff. This ensures that the sender has information that the receiver wants to know but does not know.

The sender maintains a stochastic signaling scheme $\varphi_{\eta} : S \times O \to \Delta(\Sigma)$, where $\varphi$ is parameterized by $\eta$ and $\Delta(X)$ denotes the set of all random variables on $X$. The receiver's stochastic action policy is denoted as $\pi_{\theta} : O \times \Sigma \to \Delta(A)$, where $A$ is the receiver's action space and $\theta \in \Theta$ is the policy parameter. Without loss of generality, assume that the sender takes no environmental action. The state

transition function $p : S \times A \to \Delta(S)$ and the reward functions $R^i : S \times A \to \mathbb{R}$ ($R^j : S \times A \to \mathbb{R}$) are dependent on $j$'s chosen action and are not dependent on $i$'s message. In this way, $i$ needs to influence $j$ through its signaling scheme, which indirectly affects its long-term payoff expectation.

Then, a Markov signaling game (MSG, Figure 1) is defined as a tuple

$$\mathcal{G} = \left( i, j, S, O, \Sigma, A, R^i, R^j, p, q \right).$$

At each timestep $t$ in $\mathcal{G}$, $i$ observes a state $s_t \in S$ and $j$'s observation $o_t \in O$ is sampled by $q$, and then $i$ sends a message $\sigma_t$ based on $\varphi_\eta$. Then $j$ takes action $a_t \in A$ based on its policy $\pi_\theta$ and the environment transits to the next state $s' \in S$ according to the transition function $p$. Meanwhile, player $i$ (respectively, $j$) receives the reward $r_t^i$ ($r_t^j$) via the reward function $R^i$ ($R^j$). The agents and the environment repeat this process until the environment terminates the episode.

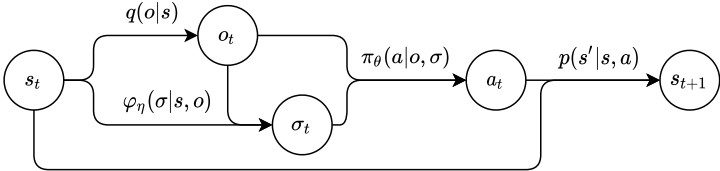

Figure 1: Illustration of the Markov signaling game. The arrows symbolize probability distributions, whereas the nodes denote the sampled variables.

The value functions in MSGs are similar to the value functions in Markov decision processes (MDPs). The sender's state value function $V_{\varphi,\pi}^i(s) = \mathbb{E}_{\varphi,\pi}\left[ G_t^i \mid s_t = s \right]$ defines the expected return of the sender at a state, where $G_t^i = \sum_{k=t}^\infty \gamma^{k-t} r_{k+1}^i$. The signal value function $Q$ for player $i$ denotes the same expectation but under an additional condition of sending signal $\sigma$ at the state $s$, as $Q_{\varphi,\pi}^i(s, \sigma) = \mathbb{E}_{\varphi,\pi}\left[ G_t^i \mid s_t = s, \sigma_t = \sigma \right]$. The action value function $U$ is the same expectation that conditions on all variables $s_t, \sigma_t, a_t$, as $U_{\varphi,\pi}^i(s, \sigma, a) = \mathbb{E}_{\varphi,\pi}\left[ G_t^i \mid s_t = s, \sigma_t = \sigma, a_t = a \right]$. Notice the transition function and the reward functions are not dependent on $\sigma$. We then define the marginal action value function $W_{\varphi,\pi}^i(s, a) = \mathbb{E}_{\varphi,\pi}\left[ G_t^i \mid s_t = s, a_t = a \right] = U_{\varphi,\pi}^i(s, \sigma, a)$. The receiver's value functions are similarly defined, by replacing $G_t^i$ with $G_t^j$.

Assume that the agents involved are self-interested, i.e., they are focusing solely on optimizing their own payoffs. However, their payoffs are general and are not necessarily the environmental rewards. For instance, the sender may replace its reward function $R^i$ with $(R^i + R^j)$ as its optimization goal, which would result in it "selfishly" optimizing for the social welfare.

MSG is new to the community and the features of MSG are quite distinctive: (1) The sender and receiver are heterogeneous, with the sender having an informational advantage; (2) Their interests may differ and thus they are in a mixed-motive scenario; (3) The sender 'signal does not influence anyone's reward. Thus the sender can only influence the receiver's belief to improve its own expected payoffs; (4) The receiver can take actions to directly determine the payoffs for both parties, but lacks sufficient information to estimate the current state and needs to obtain information from the sender.

There are several extensions of the MSGs. For example, (1) One may allow the sender to take environmental actions; (2) The sender may only have a partial observation of the state; (3) There could be multiple senders and receivers. Discussions on these extensions are deferred to Appendix B.

## 4 Method

The fundamental concept of information design is the sender optimizing its payoffs while adhering to the obedience constraints, as referenced in Equation (1). However, persuasion becomes more challenging in sequential and learning scenarios, due to the non-stationarity and the existence of the trivial equilibrium where all messages are arbitrary and are ignored.

The first obstacle is non-stationarity, a recognized issue in MARL, that is especially problematic in communication since it requires bi-level optimization. During the early stages of training, the sender's signaling scheme is mostly random and provides little information, which can quickly break the trust between mixed-motive agents and lead to the trivial equilibrium. The second difficulty is the want

of dictatorship. In the original obedience constraints, the receiver's policy is set to be deterministic by the revelation principle. It means that recommending a wrong action is non-forgiving, and thus suitable for a trial-and-error algorithm. This will also lead to the trivial equilibrium. In this section, we present how these challenges are addressed and how information design is learned.

## 4.1 Signaling Gradient

The proposed signaling gradient is utilized to compute the gradient of the sender's long-term expected payoff w.r.t. its signaling scheme parameters. When calculating this gradient, it explicitly takes into account the chain of the receiver's policy. This helps alleviate the non-stationarity between agents.

In MSGs, the signaling scheme affects the distribution of signals, indirectly impacting the receiver's action, which then determines the payoffs for all agents. Specifically, the sender's expected payoff is expanded as $\mathbb{E}_{\varphi,\pi}\left[V^i_{\varphi,\pi}(s)\right] = \sum\limits_{s,o,\sigma,a} d_{\varphi,\pi}(s) \cdot q(o \mid s) \cdot \varphi_\eta(\sigma \mid s, o) \cdot \pi_\theta(a \mid o, \sigma) \cdot U^i_{\varphi,\pi}(s, \sigma, a)$, where $d_{\varphi,\pi}$ is the state visitation frequency given $\varphi_\eta$ and $\pi_\theta$. Similar to the case of the policy gradient (PG), the relationship between state visitation frequency and $\pi$ cannot be explicitly written. The introduction of communication $\varphi$ further complicates this process as it involves deriving $\nabla_\eta d_{\varphi,\pi}(s)$.

We utilized a method similar to the policy gradient to derive the gradient $\nabla_\eta \mathbb{E}_{\varphi,\pi}\left[V^i_{\varphi,\pi}(s)\right]$ and obtain an unbiased gradient estimation. We call this estimation the *signaling gradient*. The proof of the lemma is deferred to Appendix D.

**Lemma 4.1.** *Given a signaling scheme $\varphi_\eta$ of the sender and an action policy $\pi_\theta$ of the receiver in an MSG $\mathcal{G}$, the gradient of the sender's value function $V^i_{\varphi,\pi}(s)$ w.r.t. the signaling parameter $\eta$ is*

$$\nabla_\eta V^i_{\varphi,\pi}(s) \propto \mathbb{E}_{\varphi,\pi}\left[W^i_{\varphi,\pi}(s, a) \cdot \left[\nabla_\eta \log \pi_\theta(a \mid o, \sigma) + \nabla_\eta \log \varphi_\eta(\sigma \mid s, o)\right]\right]. \tag{2}$$

It is worth noting that $\mathbb{E}\left[W^i_{\varphi,\pi}(s, a)\nabla_\eta \log \pi_\theta(a \mid o, \sigma)\right] \neq 0$ and $W^i_{\varphi,\pi}(s, a)$ takes an action as an input. As a consequence of this derivation, the sender's updated term includes the receiver's policy and action. This result aligns with intuition, as this additional term reflects the sender's consideration of its impact on the receiver.

One may naturally regard the signal as an action and directly apply the policy gradient. If so, one will obtain $\mathbb{E}_{\varphi,\pi}\left[Q^i_{\varphi,\pi}(s, \sigma) \cdot \nabla_\eta \log \varphi_\eta(\sigma \mid s)\right]$. This gradient will then be independent of the receiver's action and is therefore biased.

**Connections to Other MARL Methods** There are three perspectives to gain insights from the derivation of the signaling gradient. The first perspective is that the signaling gradient can be regarded as policy-based feedback from the receiver instead of value-based feedback in DIAL [16]. The second perspective is that the signaling gradient and LOLA [17] are similar in alleviating the non-stationarity in MARL communication. (More discussions are deferred to Appendix H.4.) Since MSGs consider the coupling decision-making processes of both parties, the signaling gradient involves the sender taking the initiative to consider how to influence the receiver. In contrast, LOLA involves an agent proactively adapting to other people's updates. The third perspective is that the signaling gradient can be seen as a first-order gradient that is absent in LIO [57]. This new gradient chain can be explicitly derived because the signaling scheme directly affects the receiver's sampling phase.

## 4.2 Policy Gradient for the Receiver

From the receiver's perspective, the decision process can be modeled as a partially observable MDP (POMDP), in which its observation is $O \times \Sigma$ in the corresponding MSG. Therefore, the receiver can optimize its payoff expectation $V^j_{\varphi,\pi}(o, \sigma)$ by calculating the gradient $\mathbb{E}_{\varphi,\pi}\left[A^j_{\varphi,\pi}(o, \sigma, a) \cdot \nabla_\theta \log \pi_\theta(a \mid o, \sigma)\right]$, where $A(o, \sigma, a) = Q(o, \sigma, a) - V(o, \sigma)$ is the advantage function [51]. Compared to the receiver in Bayesian persuasion ([28], see Section 4.5.1 for more details), the policy of the receiver in MSGs is now stochastic rather than deterministic, which allows a larger capacity in taking different actions and a larger capacity in decoding the received information.

## 4.3 Extended Obedience Constraints

In the learning algorithm context, we consider the sender's informational advantage on the current state $s$ and investigate the incentive compatibility of the receiver. As an analogous of (1), the prior

of such information is then the occupancy measure $d_{\varphi,\pi}(s)$ of the state condition on the current signaling scheme and action policy. The payoff function $w^j(s, a)$ corresponds to the action value function $W^j_{\varphi,\pi}(s, a)$ in Markov signaling games.

It amounts to deciding the signal space $\Sigma$. The revelation principle states that there is an optimal signaling scheme that does not require more signals than the number of actions available to the receiver. If one follows the revelation principle, one would reasonably use $\Sigma = A$, resulting in the following obedience constraints.

$$\sum_s d_{\varphi,\pi}(s) \cdot \varphi_\eta(a \mid s) \cdot \left[ W^j_{\varphi,\pi}(s, a) - W^j_{\varphi,\pi}(s, a') \right] \geq 0, \quad \forall a, a' \in A. \tag{3}$$

These constraints are technically correct. However, in the learning context, having only $|A|$ possible signals means that the receiver either completely follows the suggested action or completely ignores the message. The former happens only if the obedience constraints are satisfied. In sequential interactions, the constraints will, of course, be violated occasionally. However, the dictatorship nature of the signaling scheme fails to sway the receiver without consistent satisfaction with the constraints.

Moreover, consider the trivial equilibrium between the sender and the receiver: The sender does not reveal useful information, and the receiver ignores the message. At this point, neither side could escape from the equilibrium alone. This means it is likely to fail once the learning algorithm converges to the trivial equilibrium. One natural choice is to send the information to the receiver instead of dictating actions. In this way, the receiver, as a learning agent, is more likely to be able to utilize the information and is more likely to respect the message.

Therefore, we extend the obedience constraints to general, continuous signal space $\Sigma$ that describes the state. A common choice is $\Sigma = S$. The following lemma asserts that the extended obedience constraints impose the same optimum as with the obedience constraints.

**Lemma 4.2.** *Given a receiver's observation o, the extended obedience constraints (4) in MSGs yield the same optimum as the obedience constraints (3).*

$$\sum_s d_{\varphi,\pi}(s) \cdot \varphi_\eta(\sigma \mid s, o) \cdot \sum_a \left[ \pi_\theta(a \mid o, \sigma) - \pi_\theta(a \mid o, \sigma') \right] \cdot W^j_{\varphi,\pi}(s, a) \geq 0, \tag{4}$$

*for all $\sigma, \sigma' \in \Sigma$.*

The lemma assumes the sender can access the receiver's policy and observation. Otherwise, the sender may use inferring methods to maintain an estimation of that (an example is [35]). This lemma is proved using the same argument in [3]. For convenience, in later sections, the left-hand side of (4) is denoted as $C_\varphi(\sigma, \sigma')$.

## 4.4 Learning Markov Signaling Games

Given a joint policy $\pi$, the self-interested sender attempts to optimize its payoff expectation in an MSG while satisfying the extended obedience constraints. This optimization problem is

$$\max_\eta \mathbb{E}_{\varphi,\pi} \left[ V^i_{\varphi,\pi}(s) \right], \quad \text{s.t.} \quad C_\varphi(\sigma, \sigma') \geq 0, \quad \forall \sigma, \sigma'. \tag{5}$$

Since we are employing a learning-based approach, it is necessary to calculate the gradient $\nabla_\eta C_\varphi(\sigma, \sigma')$. In this way, our method is model-free and does not require the prior knowledge of $P(s)$ (the occupancy measure $d_{\varphi,\pi}(s)$). Unfortunately, when calculating the gradient of the extended obedience constraints, the technique used in the signaling gradient cannot be applied to reveal the dependency of $d_{\varphi,\pi}$ on $\varphi$. Instead, the gradient is estimated using the biased sampling method as below.

$$\nabla_\eta \hat{C}_\varphi(\sigma, \sigma') = \frac{1}{T} \sum_{s_t \in \tau} \left[ \sum_a \left( \pi_\theta(a \mid o_t, \sigma) - \pi_\theta(a \mid o_t, \sigma') \right) \cdot W^j_{\varphi,\pi}(s_t, a) \cdot \nabla_\eta \varphi_\eta(\sigma \mid s_t, o_t) \right], \tag{6}$$

where $\tau$ is a sampled trajectory with $T$ timesteps, and $\sigma'$ is randomly sampled, instead of being sampled from the signaling scheme. Despite that $\sigma$ is from the data that is generated from $\varphi$, the purpose of having these $\sigma, \sigma'$ is to find the violation of the $\forall \sigma, \sigma'$ constraints. Therefore in extended obedience constraints $\nabla_\eta \pi_\theta(a \mid o, \sigma) = 0$, which is different from the signaling gradient.

There are various methods available to solve the constrained optimization problem (5) iteratively, e.g., the Lagrangian method, the dual gradient descent method (DGD) [7]. We tested both methods in the experiments and we found that the Lagrangian method has better performance. See Appendix H.5 for the details. Taking the Lagrangian method as an example, The update of the signaling scheme parameters $\eta^{(k)}$ for the $k$-th iteration is

$$\eta^{(k+1)} \leftarrow \eta^{(k)} + \nabla_\eta \mathbb{E}_{\varphi,\pi} \left[ V^i_{\varphi,\pi}(s) \right] + \sum_{\sigma,\sigma'} \lambda_{\sigma,\sigma'} \cdot \nabla_\eta \left( \hat{C}_\varphi(\sigma, \sigma') \right)^-, \tag{7}$$

where $\lambda_{\sigma,\sigma'}$ denotes the non-negative Lagrangian multipliers (predefined as hyperparameters), and $(\cdot)^- = \min\{0, \cdot\}$.

## 4.5 Discussions on Method

### 4.5.1 Lift to the Commitment Assumption

The most controversial but reasonable assumption in information design is the commitment assumption. In Bayesian persuasion (one-to-one persuasion) [28], the sender will commit to a signaling scheme first. The sender determines its signaling scheme before the games start and honestly tells it to the receiver. A justification of the commitment assumption is that, in a repeated game where a long-term sender interacts with a sequence of short-term receivers, the commitment will naturally emerge in equilibria. This is due to the sender's need to establish its reputation for credibility, which is essential for optimizing its long-term payoff expectations [45].

However, the reputation system between the receivers still needs to be well-defined, so the receivers that have previously interacted cannot convey information about the sender to the receivers that will interact later. Without a reputation system, the sender can optimize its payoff by claiming a respectful signaling scheme while taking an exploitable one. Instead, RL allows for organic and repeated interactions between senders and receivers in a given environment, more closely resembling real-world scenarios. This unique feature enables the learning framework to capture policy evolution and better replicate phenomena in human society.

### 4.5.2 Lift to the Revelation Principle

The extended obedience constraints remove the revelation principle analysis from the obedience constraints, thereby reverting the sender's behavior from "action recommending" to "signal sending". In this way, the sender's set of signals becomes redundant. The redundancy renders the signaling scheme more general and amenable to learning-based approaches. Previously, the signaling scheme required a one-to-one mapping to recommend a particular action, where recommending an undesired action can be non-forgiving. With the introduction of redundancy, the sender can now learn many-to-one mappings to refer to the wanted action distribution, which is a more lenient way for a trial-and-error method. This redundancy is similar to other areas of learning algorithms. For example, one could increase the size of the neural network beyond the information theory necessity to better encode and represent the mapping.

### 4.5.3 Far-sighted Receiver

The nature of RL determines that the receiver in MSGs is considering the cumulative reward. This lifts the commitment assumption and evolves the trustworthiness of the sender's signaling scheme. But meanwhile, the cumulative reward formulates that the receiver must be far-sighted, which is different from the common assumption of a myopic receiver in information design. In fact, a far-sighted receiver is inevitable once we lift the commitment assumption.

A far-sighted receiver is, in general, regarded hard in the literature. Gan et al. [18] prove that information design with a far-sighted receiver is NP-hard. One could intuitively see this from the `Recommendation Letter` example (See Appendix F). The HR can deliberately choose not to hire any students, even when the signaling scheme satisfies the obedience constraints, hoping to force the professor to be more honest (i.e., reveal more information) in the future. The optimality of the receiver's policy is undefined in this setting.

One way to empirically prevent this is to set an additional constraint $\int_{\sigma,\sigma'} C_\varphi(\sigma,\sigma')d\sigma d\sigma' \geq \epsilon$ apart from the obedience constraints $C_\varphi(\sigma,\sigma') \geq 0$ in Equation (5), where $\epsilon > 0$. A more substantial improvement in reward will incentivize the receiver to establish trust in the long run.

### 4.5.4 Hyper Gradient

Our approach shares similarities with LIO (discussed in Appendix A), as both methods allow agents to alter their parameters to indirectly enhance their payoff expectations by influencing others. In their cases, agents achieve influence by offering rewards to others. The main difference between rewarding and signaling is that the former solely impacts others' policy updates (as the gained rewards are used exclusively for updating). In contrast, the latter additionally affects the sample generation.

In methods to incentivize others, the gradients of the receiver's one-step policy update w.r.t. the sender's rewarding network parameters are required to capture the sender's influence on the receiver. This kind of gradient is second-order and can be viewed as a hyper-gradient.

Unlike incentive-based interventions, in communication methods, the sender's outputs are the inputs of the receiver's actor. The sender can achieve influence while generating trajectories. Hence, our primary focus is on studying the first-order gradient of the receiver's policy w.r.t. the sender's signaling parameters, i.e., $\nabla_\eta \pi_\theta(a \mid o, \sigma)$. The second-order gradients can also be computed, as shown in Equation (21) in the appendix. The effect of the hyper gradient is left for future work.

### 4.5.5 Sender's Access to Receiver

In signaling gradient, the updating of the sender's signaling scheme needs to backpropagate through the receiver's policy during training. Accessing the opponent in mixed-motive settings might sound counterintuitive at a glance, but this setting is actually reasonable and technically feasible.

In MARL algorithms, the Centralized Training with Decentralized Execution (CTDE) framework is commonly used [35]. This means that in the simulated environment, data is centralized and accessible during training, which allows us to use all the required quantities. Once the training is completed, the agent no longer has access to other agents.

From the information design perspective, we may consider the receiver as a "dummy" learning agent that is auxiliary in centralized training. Once a sender agent is trained, it finds the (coarse correlated) equilibrium that is no longer connected to specific receivers. The sender is supposed to persuade any rational, self-interested agents in subsequent interactions with them.

Technically, in communication methods, it is common for gradients to pass through other agents during training. We adopted the commonly used Gumbel-Softmax technique in the field of emergent communication research to allow for end-to-end differentiation, which is used to retain the gradients of sampled signals [24, 20].

## 5 Experiments

The method proposed in this paper is validated in `Recommendation Letter` and `Reaching Goals`. The receiver's action policy is implemented by the advantage actor-critic (A2C) [38]. Each curve in the experimental result graphs is drawn with at least 15 random seeds. All the seeds are included in the results (including those failed ones). Running 4 seeds with 2 NVIDIA GeForce RTX 3090, the longest time is `Reaching Goals` with $5 \times 5$ map, which takes up to a day.

Our designed algorithm takes the perspective of a self-interested sender. Therefore, the measure of success for our algorithm primarily depends on the sender's reward $r^i$. However, we also present the curve of $r^i + r^j$ (social welfare) in an attempt to demonstrate that even when the sender is self-interested and the signaling scheme is somewhat deceptive, it does not harm social welfare, and it might even enhance social welfare in some scenarios.

### 5.1 Recommendation Letter

`Recommendation Letter` is a classic example in information design [15, 28]. In this task, a professor will write recommendation letters for a number of graduating students, and a company's human resources department (HR) will receive the letters and decide whether to hire the students.

The professor and the HR share a prior distribution of the candidates' quality, with a probability of $1/3$ that the candidate is strong and a probability of $2/3$ that the candidate is weak. The HR does not know exactly what each student's quality is but wants to hire strong students, while the letters are the only source of information. The HR will get a reward of $1$ for hiring a strong candidate, a penalty of $-1$ for hiring a weak candidate, and a reward of $0$ for not hiring. The professor gets a $1$ reward for each hire. In this section, we focus on analyzing the experimental results, while a classic analysis of the three situations is presented in Appendix F.

In the experiments, we primarily compared the performance of policy gradient class algorithms (PG), PG class algorithms with obedience constraints (PGOC), DIAL [16], signaling gradient (SG, our ablation), and signaling gradient with obedience constraints (SGOC, our proposed method) in learning a signaling scheme. More specifically, the PG class algorithm utilized in experiments refers to A2C. Furthermore, SG also employed A2C techniques, including the use of the actor-critic framework, target critic, and advantage function (adapted to $W^i(s,a) - V^i(s)$ in MSGs). The algorithms with obedience constraints are also required to maintain an extra critic for estimating $W^j(s,a)$. We let $\varphi_\eta(\sigma \mid s, o) = \varphi_\eta(\sigma \mid s)$ and $\Sigma = \{0, 1\}$. The performance comparisons are shown in Figure 2 (a-c).

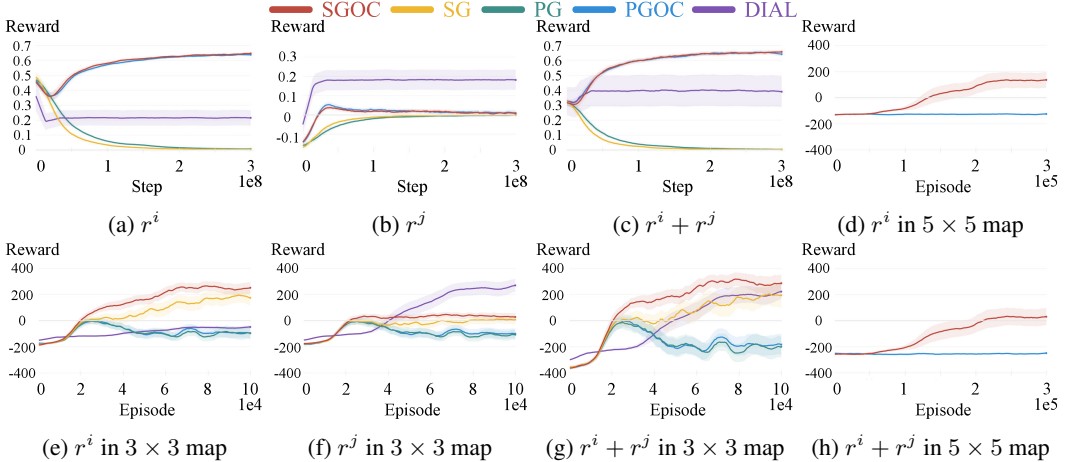

Figure 2: Comparisons of the performance. (a-c) The results of `Recommendation Letter`. (d-h) The results of `Reaching Goals`. The rewards and penalties are amplified by $20$ and $5$ ($12$ and $3.5$) respectively in $3 \times 3$ ($5 \times 5$) map.

The experiment results show that the three examples (two equilibria plus the honest signaling scheme) introduced in Section 1 have emerged. The algorithms with obedience constraints (PGOC and SGOC) reach the third equilibrium (the best for the sender), DIAL reaches the second example (the best for the receiver), and PG and SG reach the first equilibrium (the worst for both parties). Given that the `Recommendation Letter` task is not a problem with interdependent states, it is justifiable to expect that the performance of PGOC and SGOC (PG and SG) would be similar.

## 5.2 Reaching Goals

To reflect the inconsistency of the interests and the information asymmetry, we evaluate the methods in the `Reaching Goals` task. In this challenging task, the sender and receiver have different goals: the sender wants the red apple, while the receiver wants the green apple. At any given time, there is one of each type of apple on the map. Only the receiver is able to move around to reach apples in the grid-world map. The conflict of interest arises because these two types of apples are independently generated at random, and their locations are typically not the same. The receiver will not get a reward for reaching the red apple, but it does not know where the green apple is. The sender, on the other hand, knows the location of both types of apples, but can only get a reward when the receiver reaches the red apple. The sender can only indirectly optimize its payoff by sending messages to influence the receiver's actions. Moreover, this conflict can be exacerbated by designing distance penalties. After each timestep, both agents will get a penalty based on the distance between the receiver and its desired goal. And we set a fixed horizon of $50$ for each episode in this scenario. Example maps of `Reaching Goals` are shown in Figure 6.

To evaluate the efficacy of SGOC, we conducted experiments on maps of $3 \times 3$ and $5 \times 5$, and the results are presented in Figure 2. Let the signal space $\Sigma = S^1$, where $S^1$ is one channel out of three channels (the locations of two apples and the receiver's position) of the image. The receiver can only see its position. The experiments of having a larger $o^j$ are included in Appendix H.7. The smaller the $o^j$, the greater the sender's persuasion ability. From the empirical results, the signaling gradient is shown to be an essential factor in sequential communication scenarios.

### 5.3 Discussions on Experiments

#### 5.3.1 Symmetricity of the Signaling Schemes

An interesting phenomenon is observed in the `Recommendation Letter` experiments: training with different random seeds may result in different pairs of encoders and decoders. In other words, the professor may signal 1 to indicate a strong student (or recommend this student) in some seeds, while in others, this is signaled by 0. However, regardless of which case it is, the paired HR can always understand the semantics of the signals (reflected in the evolution of the receiver's policy). Based on the outcomes, the seeds are divided into two parts (A seeds and B seeds).

This phenomenon is reasonable since we do not make any prior assumption about signaling semantics. The symmetric results of `Recommendation Letter` experiments are shown in Figure 3.

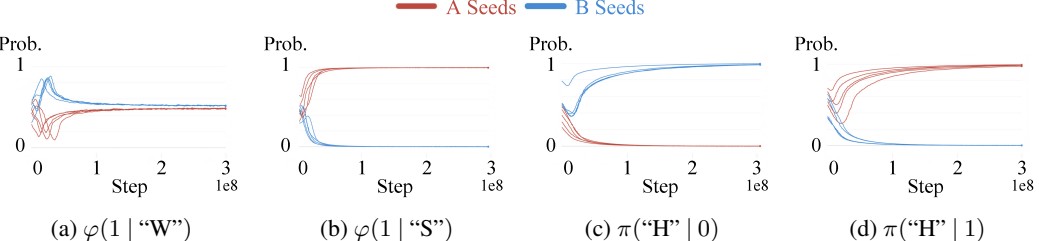

Figure 3: The signaling schemes and the action policy by SGOC. (a) The prob. of signaling 1 for **W**eak students. (b) The prob. of signaling 1 for **S**trong students. (c) The prob. of choosing to **H**ire when signaled 0. (d) The prob. of choosing to **H**ire when signaled 1.

#### 5.3.2 Honesty of the Sender

As discussed in Section 4.5.3, the obedience constraints can be associated with an additional constraint $\int_{\sigma,\sigma'} C_\varphi^j(\sigma, \sigma') d\sigma d\sigma' \geq \epsilon$. The hyperparameter $\epsilon > 0$ is set to improve the credibility of signaling schemes in practical situations and to cope with the RL capacity of the receiver. Define the honesty metric as $|\varphi(1 \mid \text{"S"}) - \varphi(1 \mid \text{"W"})|$. The experiments show that the larger the $\epsilon$ and the Lagrangian $\lambda$, the more honest the signaling scheme will be, as shown in Figure 8 in the appendix.

## 6 Conclusion

We investigate information design, a substantial and open area, for MARL that discusses mixed-motive communication. Technically, we propose the Markov signaling games to describe the problem and provide its characterizations. We then prove the signaling gradient lemma, which gives an unbiased way to estimate the gradient and update the sender's signaling network. To learn the incentive compatibility of the signaling scheme, we propose extended obedience constraints. The new constraints are more suitable for learning algorithms and practically promote mutually beneficial signaling schemes. The commitment assumption and the revelation principle are lifted by investigating information design with MARL. Experiments and extended discussions are presented to demonstrate the efficacy of our framework and algorithm.

## Acknowledgments

We extend our heartfelt gratitude to Cynthia Huang, Zongqing Lu, Shuai Li, Pascal Poupart, Dan Qiao, Han Wang, Zichuan Wan, Jiachen Yang, Zewu Zheng, Shuhui Zhu for their enlightening discussions. Part of the implementation regarding the signaling gradient was developed by Zewu Zheng.

The authors are partially supported by National Natural Science Foundation of China (62106213, 72150002), Postdoctoral Science Foundation of China (2022M723039), Shenzhen Science and Technology Program (RCBS20210609104356063, JCYJ20210324120011032), and Guangdong Provincial Key Laboratory of Big Data Computing of The Chinese University of Hong Kong, Shenzhen.

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
