(a) $\varphi(1 \mid \text{``W''})$      (b) $\varphi(1 \mid \text{``S''})$      (c) $\pi(\text{``H''} \mid 0)$      (d) $\pi(\text{``H''} \mid 1)$

Figure 3: The signaling schemes and the action policy by SGOC. (a) The prob. of signaling $1$ for **W**eak students. (b) The prob. of signaling $1$ for **S**trong students. (c) The prob. of choosing to **H**ire when signaled $0$. (d) The prob. of choosing to **H**ire when signaled $1$.

#### 5.3.2  Honesty of the Sender

As discussed in Section 4.5.3, the obedience constraints can be associated with an additional constraint $\int_{\sigma, \sigma'} C_\varphi^j(\sigma, \sigma') d\sigma d\sigma' \geq \epsilon$. The hyperparameter $\epsilon > 0$ is set to improve the credibility of signaling schemes in practical situations and to cope with the RL capacity of the receiver. Define the honesty metric as $|\varphi(1 \mid \text{``S''}) - \varphi(1 \mid \text{``W''})|$. The experiments show that the larger the $\epsilon$ and the Lagrangian $\lambda$, the more honest the signaling scheme will be, as shown in Figure 8 in the appendix.

## 6  Conclusion

We investigate information design, a substantial and open area, for MARL that discusses mixed-motive communication. Technically, we propose the Markov signaling games to describe the problem and provide its characterizations. We then prove the signaling gradient lemma, which gives an unbiased way to estimate the gradient and update the sender's signaling network. To learn the incentive compatibility of the signaling scheme, we propose extended obedience constraints. The new constraints are more suitable for learning algorithms and practically promote mutually beneficial signaling schemes. The commitment assumption and the revelation principle are lifted by investigating information design with MARL. Experiments and extended discussions are presented to demonstrate the efficacy of our framework and algorithm.

## Acknowledgments

We extend our heartfelt gratitude to Cynthia Huang, Zongqing Lu, Shuai Li, Pascal Poupart, Dan Qiao, Han Wang, Zichuan Wan, Jiachen Yang, Zewu Zheng, Shuhui Zhu for their enlightening discussions. Part of the implementation regarding the signaling gradient was developed by Zewu Zheng.

The authors are partially supported by National Natural Science Foundation of China (62106213, 72150002), Postdoctoral Science Foundation of China (2022M723039), Shenzhen Science and Technology Program (RCBS20210609104356063, JCYJ20210324120011032), and Guangdong Provincial Key Laboratory of Big Data Computing of The Chinese University of Hong Kong, Shenzhen.

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

# A    Other Related Works

**Mechanism Design**    In parallel to the setting at hand, mechanism design[3] addresses situations in which agents possess distinct and private preferences [41, 13, 47]. The crux of this approach entails devising regulations that incentivize agents to candidly disclose their preferences in their own self-interest, culminating in a collectively optimal outcome. The community has extensively investigated the analytical paradigm of mechanism design. However, this paradigm is constrained by several factors such as linear agent cost and planner incentive functions [44], finite single-stage games [34], and state-based potential games [33]. Consequently, these simplifications restrict its applicability in nonlinear, temporally-extended environments [58].

To overcome these limitations, recent works adopt the agent-based simulation [53] and utilize SOTA agent learning methods such as MARL for mechanism design. While this approach sacrifices analytical tractability, it offers greater flexibility and applicability in complex environments [58]. As a method of mechanism design, providing rewards has been applied to MARL. For example, LIO allows the RL agents to send rewards directly to others, which can be used to solve first-order social dilemmas (e.g. iterated prisoner's dilemma and tragedy of the commons) and improve social welfare [57, 58]. The other perspective of influencing by rewards is taxation. By adopting RL, the AI economist improves utilitarian social welfare in one-step co-adaptive learning scenarios [59].

Besides, to elicit the desired social choice (the aggregation of the preferences of all the agents), the method of mechanism design is not necessarily to be providing rewards, but also more general mechanisms, such as distribution rules of public goods. For applications, [29] proposed a method that designs mechanisms by RL for voting. Their designed mechanisms successfully won the majority vote at the human level in a public goods social dilemma.

**Sequential Information Design**    Information design has recently been extended to sequential scenarios [19]. To model the coupling decision processes of the sender and the receiver, Markov persuasion processes (MPPs) are proposed in [18], and [56][4]. [18] proved that persuading a far-sighted receiver in MPPs is NP-hard, and [56] proposed a learning method for persuading a bunch of one-shot myopic receivers in MPPs. On the other hand, [5] proposed a learning method for a sender to persuade a far-sighted receiver without knowing the prior belief. Besides, [11] proposed a variant of obedience constraints for persuading multiple receivers in sequential interactions. The studies above have provided a solid theoretical foundation. However, all current discussions on this topic still rely on the commitment assumption and the revelation principle and need more algorithms that work in practical scenarios.

Compared to applying mechanism design to reinforcement learning, applying information design approaches to reinforcement learning presents a more challenging task. This is because the signals directly influence the agents' interactions, not only in their update phase but also in the generation of trajectory data. In contrast, the reward in mechanism design only affects the agent's update phase since it is solely used during updates. Therefore, the existing mechanism design in MARL methods cannot be directly applied to the case of information design, and alternative analyses are required.

**Reference Games**    Reference games are usually cooperative tasks (e.g. Lewis signaling games [32]). The core of these tasks is how the sender efficiently conveys its information to the receiver, enabling the receiver to take a specific action desired by both the sender and receiver. The crucial aspect lies in how the sender conveys information to enable the receiver to understand the intended semantics. Therefore, the sender aims to transmit as much of its available information as possible. While information design focuses on mixed-motive communication tasks. The receiver's action will determine the payoffs for both parties, but there may not be a specific action that is desired by both. In fact, the self-interested sender's goal is not to maximize the transmission of its known information, but to leverage its informational advantage to confuse the self-interested receiver, so as to maximize

---

[3]In the literature, the term *mechanism design* sometimes refers to a border definition that includes direct influence methods (e.g. providing tangible goods and information), and indirect methods (e.g. building reputations systems and infrastructure systems).

[4]Although both works name their models as MPPs, they are different models. In [18], the sender's informational advantage is separated from the states and has no impact on the state transitions. In [56], a new myopic receiver interacts with the sender every timestep. Both are different from our model.

its own payoffs. The sender aims to persuade the receiver to take actions that are advantageous to the sender. In fact, senders typically learn deceptive strategies.

**Emergent Communication** Another domain closely related to our work is emergent communication [12, 10, 20, 55]. A close connection is found in [10], as they mention a negotiation environment which is indeed a mixed-motive task. The communication protocol we study employs the linguistic channel they mentioned, satisfying two primary characteristics: **(1)** Non-bindingness: Messages conveyed through this channel don't bind the sender to any action or state, given the sender's empty action space, rendering the communication as purely cheap talk; **(2)** Unverifiability: The absence of an intrinsic connection between the linguistic expression and the actual state of affairs implies a potential for deceit by the sender. Contrary to their methodology where self-interested agents fall short in achieving favorable outcomes via a linguistic channel, our approach facilitates good performance in Markov signaling games.

## B    Extensions of Markov Signaling Games

The MSG defined above can be extended to more general settings. Some of these extended models are compatible with our methods, requiring only minor modifications. Some extensions, though, require further investigation and are left for future work.

**The Sender's Actions** It is immediate to allow the sender to take action at the same time as sending signals. In cases where the sender is permitted to take environmental actions beyond signaling, the sender $i$ chooses actions $a^i \in A^i$ according to its policy $\pi^i_{\theta^i} : S \times \Sigma \to \Delta(A^i)$. Notably, the sender's action policy considers the signals it sends to the receiver in the same round. This is necessary to enable the adaptation to a variety of receiver responses induced by the dispatched signals.

**Partial Observability of the Sender** In some scenarios, the sender may only have access to a partial observation $o^i$. Under an informational advantage ($o^i - o^j \neq \varnothing$, and $\{o^i_t - o^j_t\}_{t \geq 0}$ affects $j$'s payoff) condition, there are 4 possible cases for $o^i_t$ and $o^j_t$, as shown in Figure 4. Case 2 can be

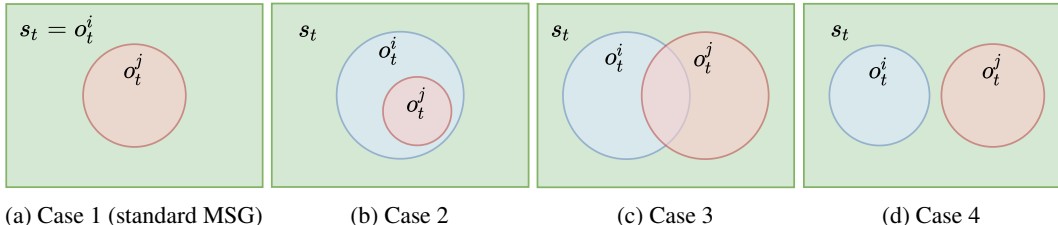

(a) Case 1 (standard MSG)    (b) Case 2    (c) Case 3    (d) Case 4

Figure 4: Sender $i$ has informational advantage over receiver $j$. The informational advantage is reflected by $o^i_t - o^j_t$. The receiver's observation in standard MSGs is turned to $o^i_t \cap o^j_t$ in every case.

handled well in practice. The sender could estimate the current state from its observation sequence, similar to methods in partially observable Markov decision processes (POMDPs) [14]. In Cases 3 and 4, the sender needs to estimate $o^j_t - o^i_t \neq \varnothing$ (information that the receiver know but the sender does not) and consider the effect of it. We leave Cases 3 and 4 for future work.

**Multiple Senders** One possible approach to model multiple senders is by modeling a separate MSG for each sender. However, it treats each sender independently and overlooks the interplay among them. The general setting should model this as a game among senders and further specify the receiver's decision based on multiple signals received from different senders. This is an important topic for future works because, more generally, more than one agent possesses such information.

**Multiple Receivers** Markov signaling games with multiple receivers are an immediate extension. The conclusions of the signaling gradient Lemma 4.1 can also be applied. However, if the sender wants to persuade a group of receivers simultaneously, additional considerations need to be taken into account for extended obedience constraints. Since the reward is determined by all receivers' actions, the sender must simultaneously consider the effects of signaling to all receivers, the dependencies of

which are not yet clear. This issue is left for future work. In this section, we redefine the model for clarity. Subsequent discussions in the appendix are based on this extension.

Consider a signaling game involving 1 sender and $N$ receivers, where $J = \{0, 1, \ldots, N-1\}$ denotes the set of receivers. The signaling channel between the sender and each receiver is private, so the messages sent through each channel are only observable to the sender and the corresponding receiver. $\Sigma^j$ denotes the message set of receiver $j \in J$, and the joint message set is defined as $\boldsymbol{\Sigma} = \prod_j \Sigma^j$. Each receiver $j \in J$ makes decisions based only on received messages and its observation $o^j \in O^j$. At each timestep, the environment generates a joint observation $\boldsymbol{o}_t \in \prod_j O^j$ according to the emission function $q : S \to \prod_j O^j$, where each signal $o_t^j \subset s_t$ is a proper subset of $s_t$. The sender's informational advantage over receiver $j$ is reflected by $s_t - o_t^j$, where $\{s_t - o_t^j\}_{t \geq 0}$ affects $j$'s payoff.

The sender maintains a stochastic signaling scheme $\varphi_\eta : S \to \Delta(\boldsymbol{\Sigma})$. The stochastic action policy of receiver $j$ is denoted as $\pi_{\theta^j}^j : O^j \times \Sigma^j \to \Delta(A^j)$. Specifically, $\pi_{\theta^j}^j(a^j \mid o^j, \sigma^j)$ represents the probability of receiver $j$ choosing an action $a^j$ given the message $\sigma^j$ and the observation $o^j$ received. The joint action space is then defined as $\boldsymbol{A} = \prod_j A^j$, where $A^j$ is the action space of receiver $j$ and $\theta^j \in \Theta^j$ is the corresponding policy parameter. And the joint policy of all agents is defined as $\boldsymbol{\pi}_{\boldsymbol{\theta}}(\boldsymbol{a} \mid \boldsymbol{o}, \boldsymbol{\sigma}) = \prod_j \pi_{\theta^j}^j(a^j \mid o^j, \sigma^j)$, where $\boldsymbol{a} \in \boldsymbol{A}$, and $\boldsymbol{\theta} \in \prod_j \Theta^j$. When the context is clear, we will drop the subscripts for the parameters and let $\pi_\theta^j$ denote $\pi_{\theta^j}^j$.

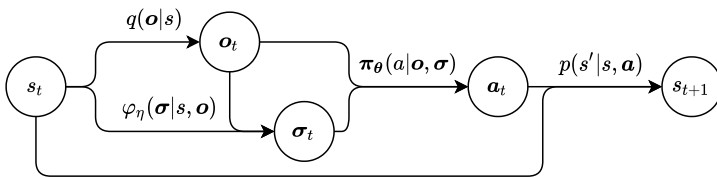

Figure 5: An illustration of the Markov signaling game with multiple receivers. The arrows symbolize probability distributions, whereas the nodes denote the sampled variables.

Then, a Markov signaling game with multiple receivers(Figure 5) is defined as a tuple

$$\mathcal{G}' = \left( i, J, S, \{O^j\}_{j \in J}, \{\Sigma^j\}_{j \in J}, \{A^j\}_{j \in J}, R^i, \{R^j\}_{j \in J}, p, q \right).$$

In $\mathcal{G}'$, the sender $i$ observes a state $s \in S$ and sends messages $\boldsymbol{\sigma}$ based on $\varphi_\eta$, and the environment generates joint observations $\boldsymbol{o}$ according to the emission function $q$. Then all agents take actions $\boldsymbol{a}$ based on the joint policy $\boldsymbol{\pi}_{\boldsymbol{\theta}}$ and the environment transits to the next state $s'$ according to the transition function $p : S \times \boldsymbol{A} \to \Delta(S)$. Meanwhile, the sender $i$ (respectively, a receiver $j$) receives the reward $r^i$ ($r^j$) via the reward function $R^i : S \times \boldsymbol{A} \to \mathbb{R}$ ($R^j : S \times \boldsymbol{A} \to \mathbb{R}$). The agents and the environment repeat this process until the environment terminates the episode.

The definition of value functions for the sender's signaling process in MSGs with multiple receivers can be immediately obtained from the definition in MSGs. The sender's state value function is defined as $V_{\varphi,\boldsymbol{\pi}}^i(s) = \mathbb{E}_{\varphi,\boldsymbol{\pi}} \left[ G_t^i \mid s_t = s \right]$. The signal value function $Q$ for the sender of taking signal $\sigma$ at the state $s$ is $Q_{\varphi,\boldsymbol{\pi}}^i(s, \boldsymbol{\sigma}) = \mathbb{E}_{\varphi,\boldsymbol{\pi}} \left[ G_t^i \mid s_t = s, \boldsymbol{\sigma}_t = \boldsymbol{\sigma} \right]$. The action value function $U$ is defined as $U_{\varphi,\boldsymbol{\pi}}^i(s, \boldsymbol{\sigma}, \boldsymbol{a}) = \mathbb{E}_{\varphi,\boldsymbol{\pi}} \left[ G_t^i \mid s_t = s, \boldsymbol{\sigma}_t = \boldsymbol{\sigma}, \boldsymbol{a}_t = \boldsymbol{a} \right]$. And still, $W_{\varphi,\boldsymbol{\pi}}^i(s, \boldsymbol{a}) = U_{\varphi,\boldsymbol{\pi}}^i(s, \boldsymbol{\sigma}, \boldsymbol{a})$.

## C Bellman Equations in Markov Signaling Games

According to the definitions of the value functions in MSGs (defined in Section 3) and the law of total expectation, it can immediately derive a variant of Bellman equations as

$$\begin{aligned} V_{\varphi,\boldsymbol{\pi}}^i(s) &= \sum_{\boldsymbol{o}} \Pr(\boldsymbol{o} \mid s) \sum_{\boldsymbol{\sigma}} \Pr(\boldsymbol{\sigma} \mid s, \boldsymbol{o}) \sum_{\boldsymbol{a}} \Pr(\boldsymbol{a} \mid s, \boldsymbol{o}, \boldsymbol{\sigma}) \cdot U_{\varphi,\boldsymbol{\pi}}^i(s, \boldsymbol{\sigma}, \boldsymbol{a}) \\ &= \sum_{\boldsymbol{o}} q(\boldsymbol{o} \mid s) \sum_{\boldsymbol{\sigma}} \varphi_\eta(\boldsymbol{\sigma} \mid s) \sum_{\boldsymbol{a}} \boldsymbol{\pi}_{\boldsymbol{\theta}}(\boldsymbol{a} \mid \boldsymbol{o}, \boldsymbol{\sigma}) \cdot U_{\varphi,\boldsymbol{\pi}}^i(s, \boldsymbol{\sigma}, \boldsymbol{a}). \end{aligned} \tag{8}$$

In particular, in our current model, we set the environment to not give any reward to the signaling processes. And there is no cost for sending a message. Thus,

$$
\begin{aligned}
U^i_{\varphi,\boldsymbol{\pi}}(s,\boldsymbol{\sigma},\boldsymbol{a}) &= \mathbb{E}_{\varphi,\boldsymbol{\pi}}\left[G^i_t \mid s_t = s, \boldsymbol{\sigma}_t = \boldsymbol{\sigma}, \boldsymbol{a}_t = \boldsymbol{a}\right] \\
&= \mathbb{E}_{\varphi,\boldsymbol{\pi}}\left[G^i_t \mid s_t = s, \boldsymbol{a}_t = \boldsymbol{a}\right] = W^i_{\varphi,\boldsymbol{\pi}}(s,\boldsymbol{a}),
\end{aligned}
\tag{9}
$$

$$
\begin{aligned}
V^i_{\varphi,\boldsymbol{\pi}}(s) &= \mathbb{E}_{\varphi,\boldsymbol{\pi}}\left[G^i_t \mid s_t = s\right] = \mathbb{E}_{\varphi,\boldsymbol{\pi}}\left[r^i_{t+1} + \gamma \cdot G^i_{t+1} \mid s_t = s\right] \\
&= \mathbb{E}_{\varphi,\boldsymbol{\pi}}\left[r^i_{t+1} + \gamma \cdot V^i_{\varphi,\boldsymbol{\pi}}(s_{t+1}) \mid s_t = s\right],
\end{aligned}
\tag{10}
$$

and

$$
\begin{aligned}
U^i_{\varphi,\boldsymbol{\pi}}(s,\boldsymbol{\sigma},\boldsymbol{a}) &= \mathbb{E}_{\varphi,\boldsymbol{\pi}}\left[G^i_t \mid s_t = s, \boldsymbol{\sigma}_t = \boldsymbol{\sigma}, \boldsymbol{a}_t = \boldsymbol{a}\right] \\
&= \mathbb{E}_{\varphi,\boldsymbol{\pi}}\left[r^i_{t+1} + \gamma \cdot G^i_{t+1} \mid s_t = s, \boldsymbol{\sigma}_t = \boldsymbol{\sigma}, \boldsymbol{a}_t = \boldsymbol{a}\right] \\
&= \mathbb{E}_{\varphi,\boldsymbol{\pi}}\left[r^i_{t+1} + \gamma \cdot V^i_{\varphi,\boldsymbol{\pi}}(s_{t+1}) \mid s_t = s, \boldsymbol{\sigma}_t = \boldsymbol{\sigma}, \boldsymbol{a}_t = \boldsymbol{a}\right] \\
&= R^i(s,\boldsymbol{a}) + \gamma \sum_{s'} p(s' \mid s, \boldsymbol{a}) \cdot V^i_{\varphi,\boldsymbol{\pi}}(s').
\end{aligned}
\tag{11}
$$

## D   Proof of the Signaling Gradient Lemma

Firstly, let $\mathrm{Pr}_{\varphi,\boldsymbol{\pi}}(s \to x, k)$ denote the probability of transferring from state $s$ to state $x$ in $k$ steps, given the signaling scheme $\varphi$ and the joint policy $\boldsymbol{\pi}$. Its recursive relationship is similar to the situation in MDPs:

$$
\mathrm{Pr}_{\varphi,\boldsymbol{\pi}}(s \to s, 0) = 1, \mathrm{Pr}_{\varphi,\boldsymbol{\pi}}(s \to s', 1) = \sum_{\boldsymbol{\sigma},\boldsymbol{o},\boldsymbol{a}} q(\boldsymbol{o} \mid s) \cdot \varphi_\eta(\boldsymbol{\sigma} \mid s) \cdot \boldsymbol{\pi}_{\boldsymbol{\theta}}(\boldsymbol{a} \mid \boldsymbol{o}, \boldsymbol{\sigma}) \cdot p(s' \mid s, \boldsymbol{a}),
$$

$$
\mathrm{Pr}_{\varphi,\boldsymbol{\pi}}(s \to x, k+1) = \sum_{s'} \mathrm{Pr}_{\varphi,\boldsymbol{\pi}}(s \to s', k) \cdot \mathrm{Pr}_{\varphi,\boldsymbol{\pi}}(s' \to x, 1).
\tag{12}
$$

Then according to 8,

$$
\nabla_\eta V^i_{\varphi,\boldsymbol{\pi}}(s) = \nabla_\eta \sum_{\boldsymbol{o}} q(\boldsymbol{o} \mid s) \sum_{\boldsymbol{\sigma},\boldsymbol{a}} \varphi_\eta(\boldsymbol{\sigma} \mid s) \cdot \boldsymbol{\pi}_{\boldsymbol{\theta}}(\boldsymbol{a} \mid \boldsymbol{o}, \boldsymbol{\sigma}) \cdot U^i_{\varphi,\boldsymbol{\pi}}(s,\boldsymbol{\sigma},\boldsymbol{a})
\tag{13}
$$

$$
= \sum_{\boldsymbol{o}} q(\boldsymbol{o} \mid s) \sum_{\boldsymbol{\sigma},\boldsymbol{a}} \nabla_\eta\left[\varphi_\eta(\boldsymbol{\sigma} \mid s) \cdot \boldsymbol{\pi}_{\boldsymbol{\theta}}(\boldsymbol{a} \mid \boldsymbol{o}, \boldsymbol{\sigma})\right] \cdot U^i_{\varphi,\boldsymbol{\pi}}(s,\boldsymbol{\sigma},\boldsymbol{a})
\tag{14}
$$

$$
+ \sum_{\boldsymbol{o}} q(\boldsymbol{o} \mid s) \sum_{\boldsymbol{\sigma},\boldsymbol{a}} \varphi_\eta(\boldsymbol{\sigma} \mid s) \cdot \boldsymbol{\pi}_{\boldsymbol{\theta}}(\boldsymbol{a} \mid \boldsymbol{o}, \boldsymbol{\sigma}) \cdot \nabla_\eta U^i_{\varphi,\boldsymbol{\pi}}(s,\boldsymbol{\sigma},\boldsymbol{a}).
\tag{15}
$$

In the following many steps of derivation, we will expand 15, leaving 14 unchanged. Since 14 is a function of the state $s$, for brevity, we will let $f_{\varphi,\boldsymbol{\pi}}(s)$ denote it:

$$
\nabla_\eta V^i_{\varphi,\boldsymbol{\pi}}(s) = f_{\varphi,\boldsymbol{\pi}}(s) + \sum_{\boldsymbol{o}} q(\boldsymbol{o} \mid s) \sum_{\boldsymbol{\sigma},\boldsymbol{a}} \varphi_\eta(\boldsymbol{\sigma} \mid s) \cdot \boldsymbol{\pi}_{\boldsymbol{\theta}}(\boldsymbol{a} \mid \boldsymbol{o}, \boldsymbol{\sigma}) \cdot \nabla_\eta U^i_{\varphi,\boldsymbol{\pi}}(s,\boldsymbol{\sigma},\boldsymbol{a})
$$

$$
= f_{\varphi,\boldsymbol{\pi}}(s) + \sum_{\boldsymbol{o}} q(\boldsymbol{o} \mid s) \sum_{\boldsymbol{\sigma},\boldsymbol{a}} \varphi_\eta(\boldsymbol{\sigma} \mid s) \cdot \boldsymbol{\pi}_{\boldsymbol{\theta}}(\boldsymbol{a} \mid \boldsymbol{o}, \boldsymbol{\sigma}) \cdot \nabla_\eta\left[R^i(s,\boldsymbol{a}) + \gamma \sum_{s'} p(s' \mid s, \boldsymbol{a}) \cdot V^i_{\varphi,\boldsymbol{\pi}}(s')\right]
$$

$$
= f_{\varphi,\boldsymbol{\pi}}(s) + \gamma \sum_{\boldsymbol{o}} q(\boldsymbol{o} \mid s) \sum_{\boldsymbol{\sigma},\boldsymbol{a}} \varphi_\eta(\boldsymbol{\sigma} \mid s) \cdot \boldsymbol{\pi}_{\boldsymbol{\theta}}(\boldsymbol{a} \mid \boldsymbol{o}, \boldsymbol{\sigma}) \cdot \sum_{s'} p(s' \mid s, \boldsymbol{a}) \cdot \nabla_\eta V^i_{\varphi,\boldsymbol{\pi}}(s')
$$

$$
= f_{\varphi,\boldsymbol{\pi}}(s) + \gamma \sum_{s'} \mathrm{Pr}_{\varphi,\boldsymbol{\pi}}(s \to s', 1) \cdot \nabla_\eta V^i_{\varphi,\boldsymbol{\pi}}(s').
$$

$$
\tag{16}
$$

Keep unrolling recursively,

$$\nabla_\eta V^i_{\varphi,\boldsymbol{\pi}}(s) = f_{\varphi,\boldsymbol{\pi}}(s) + \gamma \sum_{s'} \mathrm{Pr}_{\varphi,\boldsymbol{\pi}}(s \to s', 1) \cdot \nabla_\eta V^i_{\varphi,\boldsymbol{\pi}}(s')$$

$$= f_{\varphi,\boldsymbol{\pi}}(s) + \gamma \sum_{s'} \mathrm{Pr}_{\varphi,\boldsymbol{\pi}}(s \to s', 1) \cdot \left[ f_{\varphi,\boldsymbol{\pi}}(s') + \gamma \sum_{s''} \mathrm{Pr}_{\varphi,\boldsymbol{\pi}}(s' \to s'', 1) \cdot \nabla_\eta V^i_{\varphi,\boldsymbol{\pi}}(s'') \right]$$

$$= f_{\varphi,\boldsymbol{\pi}}(s) + \gamma \sum_{s'} \mathrm{Pr}_{\varphi,\boldsymbol{\pi}}(s \to s', 1) \cdot f_{\varphi,\boldsymbol{\pi}}(s') + \gamma^2 \sum_{s''} \mathrm{Pr}_{\varphi,\boldsymbol{\pi}}(s \to s'', 2) \cdot \nabla_\eta V^i_{\varphi,\boldsymbol{\pi}}(s'') \quad (17)$$

$$\cdots$$

$$= \sum_{x \in S} \sum_{k=0}^{\infty} \gamma^k \cdot \mathrm{Pr}_{\varphi,\boldsymbol{\pi}}(s \to x, k) \cdot f_{\varphi,\boldsymbol{\pi}}(x).$$

Let $h_{\varphi,\boldsymbol{\pi}}(x)$ denote $\sum_{k=0}^{\infty} \gamma^k \cdot \mathrm{Pr}_{\varphi,\boldsymbol{\pi}}(s \to x, k)$. Then the stationary distribution $d_{\varphi,\boldsymbol{\pi}}(s)$ is defined as

$$d_{\varphi,\boldsymbol{\pi}}(s) = \frac{h_{\varphi,\boldsymbol{\pi}}(s)}{\sum_{x \in S} h_{\varphi,\boldsymbol{\pi}}(x)}. \quad (18)$$

Considering the objective of signaling gradient is to optimize $V^i_{\varphi,\boldsymbol{\pi}}(s_0)$, then we have

$$\nabla_\eta V^i_{\varphi,\boldsymbol{\pi}}(s_0) = \sum_s h_{\varphi,\boldsymbol{\pi}}(s) \cdot f_{\varphi,\boldsymbol{\pi}}(s) = \left( \sum_s h_{\varphi,\boldsymbol{\pi}}(s) \right) \sum_s \frac{h_{\varphi,\boldsymbol{\pi}}(s)}{\sum_s h_{\varphi,\boldsymbol{\pi}}(s)} \cdot f_{\varphi,\boldsymbol{\pi}}(s)$$

$$\propto \sum_s \frac{h_{\varphi,\boldsymbol{\pi}}(s)}{\sum_s h_{\varphi,\boldsymbol{\pi}}(s)} \cdot f_{\varphi,\boldsymbol{\pi}}(s) = \sum_s d_{\varphi,\boldsymbol{\pi}}(s) \cdot f_{\varphi,\boldsymbol{\pi}}(s)$$

$$= \sum_s d_{\varphi,\boldsymbol{\pi}}(s) \cdot \sum_{\boldsymbol{o}} q(\boldsymbol{o} \mid s) \sum_{\boldsymbol{\sigma},\boldsymbol{a}} U^i_{\varphi,\boldsymbol{\pi}}(s, \boldsymbol{\sigma}, \boldsymbol{a}) \cdot \nabla_\eta \left[ \varphi_\eta(\boldsymbol{\sigma} \mid s) \cdot \boldsymbol{\pi}_\theta(\boldsymbol{a} \mid \boldsymbol{o}, \boldsymbol{\sigma}) \right] \quad (19)$$

$$= \sum_{s,\boldsymbol{o},\boldsymbol{\sigma},\boldsymbol{a}} d_{\varphi,\boldsymbol{\pi}}(s) \cdot q(\boldsymbol{o} \mid s) \cdot U^i_{\varphi,\boldsymbol{\pi}}(s, \boldsymbol{\sigma}, \boldsymbol{a}) \cdot \varphi_\eta(\boldsymbol{\sigma} \mid s) \cdot \boldsymbol{\pi}_\theta(\boldsymbol{a} \mid \boldsymbol{o}, \boldsymbol{\sigma}) \cdot \frac{\nabla_\eta \varphi_\eta(\boldsymbol{\sigma} \mid s)}{\varphi_\eta(\boldsymbol{\sigma} \mid s)}$$

$$+ \sum_{s,\boldsymbol{o},\boldsymbol{\sigma},\boldsymbol{a}} d_{\varphi,\boldsymbol{\pi}}(s) \cdot q(\boldsymbol{o} \mid s) \cdot U^i_{\varphi,\boldsymbol{\pi}}(s, \boldsymbol{\sigma}, \boldsymbol{a}) \cdot \varphi_\eta(\boldsymbol{\sigma} \mid s) \cdot \boldsymbol{\pi}_\theta(\boldsymbol{a} \mid \boldsymbol{o}, \boldsymbol{\sigma}) \cdot \frac{\nabla_\eta \boldsymbol{\pi}_\theta(\boldsymbol{a} \mid \boldsymbol{o}, \boldsymbol{\sigma})}{\boldsymbol{\pi}_\theta(\boldsymbol{a} \mid \boldsymbol{o}, \boldsymbol{\sigma})}$$

$$= \mathbb{E}_{\varphi,\boldsymbol{\pi}} \left[ U^i_{\varphi,\boldsymbol{\pi}}(s, \boldsymbol{\sigma}, \boldsymbol{a}) \cdot \left[ \nabla_\eta \log \varphi_\eta(\boldsymbol{\sigma} \mid s) + \nabla_\eta \log \boldsymbol{\pi}_\theta(\boldsymbol{a} \mid \boldsymbol{o}, \boldsymbol{\sigma}) \right] \right].$$

Finally, by substituting $U^i_{\varphi,\boldsymbol{\pi}}(s, \boldsymbol{\sigma}, \boldsymbol{a})$ by $W^i_{\varphi,\boldsymbol{\pi}}(s, \boldsymbol{a})$ (as analyzed in 9), the deriving result of the signaling gradient is

$$\nabla_\eta V^i_{\varphi,\boldsymbol{\pi}}(s_0) = \mathbb{E}_{\varphi,\boldsymbol{\pi}} \left[ W^i_{\varphi,\boldsymbol{\pi}}(s, \boldsymbol{a}) \cdot \left[ \nabla_\eta \log \varphi_\eta(\boldsymbol{\sigma} \mid s) + \nabla_\eta \log \boldsymbol{\pi}_\theta(\boldsymbol{a} \mid \boldsymbol{o}, \boldsymbol{\sigma}) \right] \right]. \quad (20)$$

# E  Hyper Gradient in the Signaling Gradient

Note that in every $\nabla_\eta \log \pi^j_\theta(a^j \mid o^j, \sigma^j)$, it can be decomposed to two parts:

$$\nabla_\eta \log \pi^j_\theta(a^j \mid o^j, \sigma^j) = \frac{\partial \log \pi^j_\theta(a^j \mid o^j, \sigma^j)}{\partial \pi^j_\theta(a^j \mid o^j, \sigma^j)} \cdot \frac{\partial \pi^j_\theta(a^j \mid o^j, \sigma^j)}{\partial \eta}$$

$$= \frac{1}{\pi^j_\theta(a^j \mid o^j, \sigma^j)} \left[ \frac{\partial \pi^j_\theta(a^j \mid o^j, \sigma^j)}{\partial \sigma^j} \cdot \frac{\partial \sigma^j}{\partial \eta} + \frac{\partial \pi^j_\theta(a^j \mid o^j, \sigma^j)}{\partial \theta^j} \cdot \frac{\partial \theta^j}{\partial \eta} \right]. \quad (21)$$

Similar to LIO, it can be considered that $\eta$ affects the update process of $\theta^j$. In this way,

$$\frac{\partial \theta^j}{\partial \eta} \approx \frac{\partial \Delta \theta^j}{\partial \eta}, \tag{22}$$

where $\Delta \theta^j$ is the difference for one-step update $\theta^j \leftarrow \theta^j + \Delta \theta^j$.

## F   Details and Analysis of Recommendation Letter

To better understand the information design problem, we illustrate it with an example of `Recommendation Letter` and its Bayesian persuasion solution [15, 28]. In the example, a professor will write recommendation letters for a number of graduating students, and a company's human resources department (HR) will receive the letters and decide whether to hire the students. The professor and the HR share a prior distribution of the candidates' quality, with a probability of $1/3$ that the candidate is strong and a probability of $2/3$ that the candidate is weak. The HR does not know exactly what each student's quality is but wants to hire strong students, while the letters are the only source of information. The HR will get a reward of 1 for hiring a strong candidate, a penalty of $-1$ for hiring a weak candidate, and a reward of 0 for not hiring. The professor gets a 1 reward for each hire. There are three types of outcomes between the professor and the HR:

- Since there are more weak students than strong students, the HR tends not to hire anyone if the professor does not write letters.

- If the professor honestly reports the qualities of the students, then the HR will accurately hire those strong candidates. Then their payoff expectations are both $1/3$;

- The professor reports the qualities of the strong students honestly and lying with a probability of $(1/2 - \epsilon)$ for weak students, for some arbitrarily small $\epsilon$. The optimal policy of HR is to respect the professor's recommendations. In this way, the professor's and the HR's payoff expectations are $(2/3 - 2\epsilon/3)$ and $2\epsilon/3$, respectively.

The critical insight from the example is that the information provider (the professor) needs to "lie" about its information to get the best interest. This lie, meanwhile, must still reveal part of the truth so that the information receiver (the HR) respects the information because it will benefit from the posterior belief in its best interest. The condition that the receiver benefits from the message is known as the *obedience constraints* in information design, which implies the incentive compatibility of the receiver. The sender must subtly balance the benefits of both parties under this condition and carefully design the information to be sent.

It is easy to analyze that HR can only make decisions based on the prior probability distribution if the professor does not write a recommendation letter, and its best policy is to refuse to hire any student. In this way, the professor and the HR payoffs are both 0, which is obviously a bad situation for both parties. If the professor tells the HR the student's quality honestly (i.e., the professor gives up its informational advantage), then the best strategy for the HR is to hire strong students and not weak students. In the case of the honest signaling scheme, the payoff expectations of the professor and the HR are $1/3$.

The professor can change its signaling scheme to make its payoff expectation higher, which is exactly the primary concern of information design. If the current student is strong, the professor will report it honestly; otherwise, the professor tells the HR that it is strong with a probability of $(1/2 - \epsilon)$, where $\epsilon \in (0, 1/2]$. When HR heard the professor say this was a weak student, it knew the student must be weak, so it would refuse to hire her. And the HR can calculate that $1/3$ of the students are strong, and the professor will call them strong, and $(1/3 - 2\epsilon/3)$ of students are weak, but the professor will call them strong still. So when the professor says that the current student is strong, the probability of being a strong student is $1/(2 - 2\epsilon)$, and the probability of being a weak student is $(1 - 2\epsilon)/(2 - 2\epsilon)$. Then, when the professor recommends the student, the payoff expectation of the HR of choosing to hire is $\epsilon/(1 - \epsilon)$, and the payoff expectation of choosing not to hire is 0. A rational HR will select the action that can maximize its payoff expectation. That is to say, when the professor says that the current student is strong, the HR will choose to hire. In this case, the payoff expectation of the professor is $(2/3 - 2\epsilon/3)$, and the payoff expectation of the HR is $2\epsilon/3$. It can be found that when epsilon takes $1/2$, the signaling scheme degenerates into the honest one.

# G   Details of Reaching Goals

The basics of `Reaching Goals` are introduced in Section 5.2. This section aims to offer more specific details. Example maps of `Reaching Goals` are shown in Figure 6.

At any given time in the map, there is only one target goal for the sender and one for the receiver, both uniformly distributed and randomly generated. Once the receiver reaches a goal, it will be regenerated. And the regenerated goal and the receiver will not be in the same position. An episode will only end when the specified step limit is reached. The receiver's actions consist of moving up, down, left, or right by one square. When the receiver's position coincides with a particular goal, the apple will be automatically harvested.

When the receiver's decision to pursue the green apple would mean moving away from the red goal. And in a fixed-length episode, this will reduce the receiver's goal harvesting efficiency. Since the respawn locations of goals are randomly and uniformly distributed, the positions of goals are highly likely to be non-coincident. As the map size increases, the conflict of interest between the sender and the receiver increases.

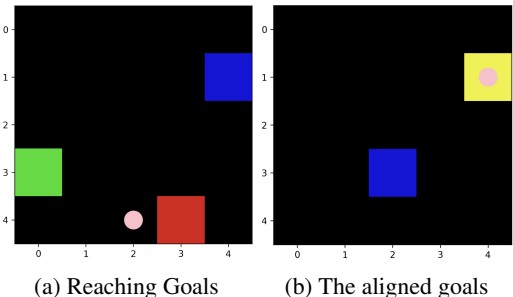

(a) Reaching Goals          (b) The aligned goals

Figure 6: Maps $5 \times 5$ of `Reaching Goals`. The blue, red, and green squares represent the receiver, the sender's goal, and the receiver's goal, respectively. If the red square and the green square overlap, it will turn yellow, meaning that the goals of agents are aligned. The pink dots represent the messages sent by the sender. The sender is out of the map.

# H   More Discussions and Results

## H.1   Incentive Compatibility of the Obedience Constraints

The obedience constraints are the core of information design and it significantly improves the performance in experiments. Assuming that the sender's signal set is equal to the receiver's action set, the sender's signals can be interpreted as recommending the receiver to take a specific action (This common assumption is without loss of generality according to the revelation principle). Under this premise, obedience constraints ensure that the receiver will definitely follow the sender's recommendations.

To explain why is it that as long as the sender's signaling scheme satisfies obedience constraints, the receiver will definitely follow the sender's recommendations. We provide a simple derivation adapted from [3] as follows:

$$\sum_s \mu_0(s) \cdot \varphi(a \mid s) \cdot \left( r^j(s, a) - r^j(s, a') \right) \geq 0$$

$$\Leftrightarrow \sum_s \frac{\mu_0(s) \cdot \varphi(a \mid s)}{\sum\limits_{s'} \mu_0(s') \cdot \varphi(a \mid s')} \cdot \left( r^j(s, a) - r^j(s, a') \right) \geq 0, \quad \forall a' \in A$$

$$\Leftrightarrow \sum_s \mu(s \mid a) \cdot \left( r^j(s, a) - r^j(s, a') \right) \geq 0, \qquad \forall a' \in A$$

$$\Leftrightarrow \sum_s \mu(s \mid a) \cdot r^j(s, a) \geq \sum_s \mu(s \mid a) \cdot r^j(s, a'), \qquad \forall a' \in A$$

where $\mu_0$ represents the prior probability, and $\mu$ represents the posterior probability. Therefore, a self-interested and rational receiver will definitely follow the sender's recommendations, because the posterior expected payoff of the action recommended by the sender is greater than or equal to the posterior expected payoffs of all other actions.

## H.2 The Non-Stationarity is Alleviated by the Signaling Gradient

This issue is a very important one and involves the insight behind signaling gradient. Non-stationarity is a common problem in the field of MARL. Each RL agent only focuses on its own learning, which means that every agent treats all quantities except itself as part of the environment, treating other agents as part of the environment. As a result, for an agent with policy, when other agents update, its environment changes, and even with the same policy, its value function will change as a result of the updates to other agents' states. We believe this phenomenon arises due to the independence of agents. However, our signaling gradient does not update its network independently; instead, it considers the impact of the receiver's policy. This is precisely the core insight behind signaling gradient.

## H.3 Discussions about DIAL

For the same experiment, if we add a constant to all sender rewards, the sender can still persuade the receiver using the same signaling scheme (as the receiver's reward remains unchanged), and thus the receiver's utility remains unaffected.

In this case, the difference between fully cooperative communication methods (e.g., DIAL) and our method becomes evident. Because DIAL only considers the receiver's reward and does not utilize the sender's reward (according to their concept, their method does not even use $r^i$ and does not reflect the self-interested sender). If we add a constant to all sender rewards, their method's social welfare remains unchanged, while our method's social welfare improves.

## H.4 Discussions about LOLA

When multiple learning agents are involved in a training process, each agent's environment becomes non-stationary. Modeling each agent as a POMDP means treating other people as part of the environment, so when other agents' policies are updated, each agent's environment changes. It can result in unstable training or undesired final results. To alleviate the non-stationarity, LOLA [17] was proposed. This method lets each agent notice the update process of others to adapt itself to this ever-changing environment. This awareness is implemented by accounting for others' gradient when an agent updates its policy. The signaling gradient we proposed is also to solve the non-stationarity problem. From the perspective of conclusion, the sender's signaling gradient also considers the receiver's policy. However, our goal is to improve the sender's ability to influence the receiver, and using LOLA would encourage the agent to adapt to updates from other agents.

Another potential question is whether LOLA can be used as a replacement for obedience constraints. LOLA is not a direct replacement for obedience constraints but rather a technique that can be used to enhance the receiver's algorithm. LOLA emphasizes an agent's consideration of others' updates during its updates to adapt to non-stationarity in multi-agent scenarios. When applied to a receiver's algorithm, LOLA allows the receiver to consider changes in the signaling scheme, which can improve its performance. On the other hand, obedience constraints restrict the sender to ensure that the signals it emits give the impression to the receiver that its rewards are not significantly reduced. Adapting the receiver to changes in signaling may not necessarily make the receiver follow the sender's recommendations. In contrast, obedience constraints provide a more muscular incentive-compatible condition regarding rewards, which can more effectively align the sender's influence.

## H.5 Discussions about the Dual Gradient Descent Method

One reasonable consideration is whether the dual gradient descent method (DGD) can be used to learn Markov signaling games as discussed in Equation (5), as this approach does not require tuning the Lagrangian multipliers. Other works, such as [43] and [23], have also utilized the DGD method.

By applying the dual gradient descent, $\eta$ and $\boldsymbol{\lambda}$ are updated as

$$\eta \leftarrow \eta - \alpha \cdot \nabla_\eta L(\eta, \boldsymbol{\lambda}), \quad \boldsymbol{\lambda} \leftarrow (\boldsymbol{\lambda} + \alpha \cdot \nabla_{\boldsymbol{\lambda}} L(\eta, \boldsymbol{\lambda}))^+ , \tag{23}$$

where $\alpha$ is the learning rate, $(\cdot)^+ = \max\{0, \cdot\}$, and $L(\eta, \boldsymbol{\lambda}) = -\mathbb{E}_{\varphi, \boldsymbol{\pi}}\left[V^i(s)\right] - \sum\limits_{j, \sigma^j, \sigma^{j'}} \lambda_{j, \boldsymbol{\sigma}, \boldsymbol{\sigma}'} \cdot$

$C_\varphi^j(\boldsymbol{\sigma}, \boldsymbol{\sigma}')$ is the Lagrangian function of Equation (5). All the gradients required have been discussed. The comparisons of performance in the `Recommendation Letter` and the `Reaching Goals` experiments are shown in Figure 7.

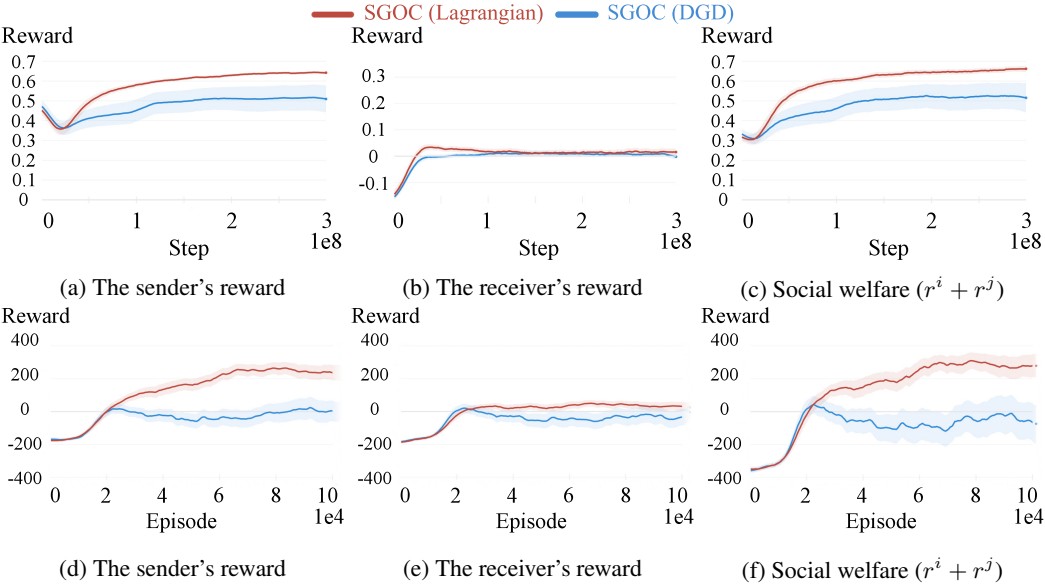

Figure 7: Performance comparisons of SGOC and DGD. (a-c) The results of `Recommendation Letter`. (d-f) The results of `Reaching Goals` with $3 \times 3$ map. The rewards and penalties are amplified by 20 and 5 respectively. The receiver can only see its position.

## H.6 Results with Different Hyperparameters

We plot the heatmap of the honesty metric against $\epsilon, \lambda$, as shown in Figure 8. We observe that the honesty metric increases with $\epsilon$ and $\lambda$, which agrees with our intuition. Specifically, when lambda reaches over 3.75 (respectively, $\epsilon$, 0.15), the honesty stops increasing, which means the sender is being very honest in this region. The best value for $\lambda$ is then somewhere between 0 to 5 (respectively, $\epsilon$, 0 to 0.3).

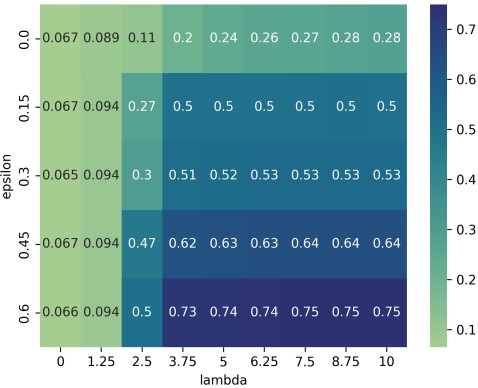

Figure 8: Honesty heatmap of the sender's signaling scheme in `Recommendation Letter`.

### H.7 Results with Different Observation of the Receiver

The algorithm proposed in this paper is suitable for scenarios where the sender has an informational advantage (discussed in Section 3). We conducted various experiments to investigate the impact of the receiver's observation in the SGOC algorithm in the `Reaching Goals` scenarios. The results, shown in Figure 9, compare the situations where the receiver "cannot see anything" (No-obs), "can only see its location" (Pos-obs), and "can see both its location and the location of its preferred apple" (Full-obs). The results indicate that the sender's payoff decreases as the receiver knows more (the sender's informational advantage decreases).

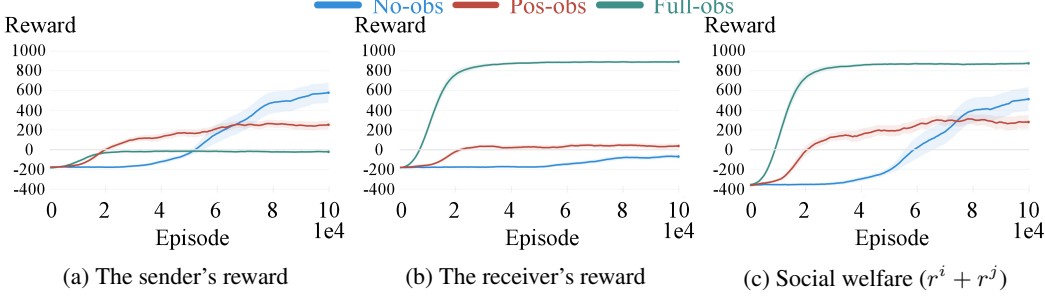

(a) The sender's reward  (b) The receiver's reward  (c) Social welfare $(r^i + r^j)$

Figure 9: Performance comparisons in `Reaching Goals` with $3 \times 3$ map. The rewards and penalties are amplified by $20$ and $5$ respectively.

## I Broader Impacts

In practical economic scenarios, persuasion is ubiquitous and plays a crucial role. As stated in the title and conclusion of McCloskey and Klamer's paper, "one quarter of the GDP is persuasion" [36]. This kind of persuasion demonstrates that communication is conceivable in mixed-motive scenarios. In this paper, we build upon the classic model of Bayesian persuasion to discuss how RL agents in sequential settings can communicate under mixed motives (most of the related research focused on fully cooperative situations).

In such a setting, the sender optimizes its own expectation while considering the incentive compatibility of the receiver, and this constraint is relatively weak. The rationality of the receiver assumes it follows the sender's instructions even if it leads to just a slightly better posterior payoff. Therefore, the receiver is in a relatively disadvantaged position in equilibrium. We discussed the potential of a far-sighted receiver to protect itself from excessive information exploitation in Section 4.5.3. Furthermore, the receiver's capability can be enhanced through the reputation mechanism in social science. Multiple receivers can evaluate the sender and refuse to cooperate with those who have a poor reputation. This compels the sender to reveal more information based on variants of obedience constraints. The insight behind this process is similar to the ultimatum game [54].

Additionally, since the optimization objective of persuasion is the sender's benefit, its equilibrium may not necessarily be advantageous for social welfare. We discussed situations where the sender can "selfishly" optimize social welfare in Section 3. Practice use of information design should take social welfare into some serious consideration.

## J Limitations and Future Works

Our current model only considers the scenario where one sender persuades one receiver. For more complex situations, additional factors need to be taken into account, which are discussed in Appendix B. The setting of multiple senders is not addressed in this manuscript. The setting of multiple receivers is formulated in the appendix, but $\varphi_\eta : S \to \Delta(\boldsymbol{\Sigma})$ has a very large signal space. If one uses $\Delta(\boldsymbol{\Sigma}) = \Delta(\Sigma_1) \times \cdots \times \Delta(\Sigma_J)$, it reduces to the one-receiver formulation. More subtle factorization of the signaling scheme is yet to be discussed. The setting of multiple receivers is yet to be empirically tested.

Our work invokes many future directions. One problem is to consider multiple senders, where the game between the sender and the receiver and between the sender and other senders co-exist. Extending the results to multiple senders will cover a wider range of real applications. Another direction is to consider the hyper gradient of the receiver's action policy concerning the sender's signaling scheme on the equilibria. This hopefully will provide a more accurate description of the learning process. Additionally, one may also use our framework to investigate a far-sighted receiver. By arming the receiver with the awareness of the sender updates, they could learn not to respect the sender, even if it is more rewarding immediately, for a better equilibrium in the long run.