# OpenReview forum: "Information Design in Multi-Agent Reinforcement Learning"
_NeurIPS.cc/2023/Conference — NeurIPS 2023 poster_

### Official Review · Reviewer_YNGH · 2023-06-27

**Soundness:** 3 good
**Presentation:** 3 good
**Contribution:** 2 fair
**Rating:** 5
**Confidence:** 3

**Summary:**

This paper studied a Markov signaling game and proposed to use the signalling gradient and extended obedience constraint to build a learning system that can effectively train the sender in the game to reach desirable equilibriums. Experiments performed on two toy problems suggest that the newly developed SGOC algorithm can perform well and reach desirable equilibriums, compared to several competing learning methods.

**Strengths:**

It is interesting and important to study the information sharing problem in a multi-agent context. The Markov signaling game presented in this paper appears to be of some practical values and deserves further investigation. The design of the new SGOC algorithm also introduces some novel ideas.

**Weaknesses:**

It is not clear whether the Markov signaling game introduced in this paper is new and has never been studied previously. The novelty of this game may need to be clarified. Its usefulness for practical applications also seems uncertain. I cannot immediately relate the two toy problems studied experimentally in this paper with any practically important scenarios where an effective machine learning-based solution of the Markov signaling game is essential. Hence, the motivation and research value of this paper may need to be significantly strengthened.

The above concern is partially triggered by the lack of discussion of related works in the main text of this paper. Because of this, the importance and novelty of studying the Markov signaling game remain doubtful to a certain extent. The true technical novelty of using the signaling gradient and the extended obedience constraint hence should be further investigated.

It is assumed that the sender can access the receiver's policy and observation (the receiver's observation is not necessarily accessible to the sender). I have no idea when such an assumption could be true in a practical scenario. In fact, the calculation of the signaling gradient may rely on this assumption. However, when the sender and receiver have potentially conflicting interests, it is entirely unclear why the receiver would be motivated to honestly support the sender by providing its internal information to the sender (in a conflicting scenario, the sender and receiver may not have the motivation to participate in a centralized training process). This further makes me question the practical value of the new SGOC algorithm.

Some statements in the paper sound a bit questionable and needs further clarification and justification. For example, the authors mentioned several times that the signaling gradient can alleviate the non-stationarity between the sender and receiver. However, it remains unknown to me why the non-stationarity can be alleviated in this way. Moreover, since the receiver is handling a partially-observable problem, it may consider to adopt a recurrent neural network as its policy network. In view of this, the calculation of the signaling gradient may become significantly more complicated. Meanwhile, the idea of biased policy gradient has been mentioned multiple times in this paper. However, I can hardly understand its true meaning. Hence, related discussions in the paper do not sound highly convincing to me.

Besides the above, I have no idea why $\nabla_{\eta}\pi_{\theta}(a|o,\sigma)$ should be 0 and why the sender's signaling scheme and the receiver's action policy should be updated alternately rather than simultaneously. This as well as several other algorithmic design details appear to have some technical issues.

The authors discussed the necessity of establishing a reputation system. However, the feasibility, complexity, and difficulty of building such a reputation system have not been investigated with sufficient depth. While building such a reputation system may not be necessary for this study, its feasibility could significantly affect the importance and practicality of the newly developed algorithm.

**Questions:**

Why is the Markov signaling game studied in this paper new and practically important?

What is the practical difficulty of supporting/realizing the commitment assumption? When the sender and receiver have potentially conflicting interests, why would the receiver be motivated to honestly support the sender by providing its internal information to the sender?

Why will the non-stationarity issue be alleviated by using signalling gradient?

What is the exact meaning of biased policy gradient?

Why should $\nabla_{\eta}\pi_{\theta}(a|o,\sigma)$ be 0? Why should the sender's signaling scheme and the receiver's action policy be updated alternately rather than simultaneously?

What are the feasibility, complexity, and difficulty of building a reputation system to support the basic assumptions of the new algorithm and the Markov signaling game?

**Limitations:**

I do not have any concerns regarding this question.

---

> ### Author Rebuttal · Authors · 2023-08-10
>
> We thank the reviewer for their insightful review. We now provide our responses.
>
> We are glad that the reviewer is interested in our work and asked many questions. Despite this, there are factual misunderstandings in the review which might have affected the evaluation of our work. We wish that with our responses and explanation, the reviewer might re-evaluate the work should they have their concerns addressed.
>
> ### Q1: Why is the Markov signaling game studied in this paper new and practically important?
>
> The reviewer mentioned a lack of related work. Due to space constraints, our related work is only mentioned in the main text for information design, while the remaining parts are placed in Appendix A. This includes discussions on mechanism design, sequential information design, and conventional MARL communication methods.
>
> ### Q2: What is the practical difficulty of supporting/realizing the commitment assumption?
>
> We believe that this is a factual misunderstanding in the review. Our proposed algorithm does not rely on the commitment assumption. On the contrary, the commitment assumption is a limitation of many information design algorithms, and one of our contributions is actually the removal of this assumption. We specifically address this issue in **section 4.5.1**.
>
> ### Q3: When the sender and receiver have potentially conflicting interests, why would the receiver be motivated to honestly support the sender by providing its internal information to the sender?
>
> Yes, it is reasonable to have the access. Please kindly refer to the global response for a detailed explanation.
>
> ### Q4: Why will the non-stationarity issue be alleviated by using signaling gradient?
>
> This issue is a very important one and involves the insight behind signaling gradient. Non-stationarity is a common problem in the field of MARL. Each RL agent only focuses on its own learning, which means that every agent treats all quantities except itself as part of the environment, treating other agents as part of the environment. As a result, for an agent with policy $\pi$, when other agents update, its environment changes, and even with the same policy, its value function will change as a result of the updates to other agents' states. We believe this phenomenon arises due to the independence of agents. However, our signaling gradient does not update its network independently; instead, it considers the impact of the receiver's policy. This is precisely the core insight behind signaling gradient.
>
> ### Q5: What is the exact meaning of biased policy gradient?
>
> Similar to Sutton's policy gradient theorem, we explicitly consider the complete Markov model that includes both the sender and receiver, and prove that the signaling gradient is unbiased. Here, the term "unbiased" has the same meaning as in the policy gradient theorem, indicating that the gradients we compute are accurate. If we were to treat the sender's signaling scheme as a regular action policy and update it using policy gradient, we would overlook a significant amount of gradient information, resulting in bias.
>
> Our signaling gradient is:
> $$ \nabla_\eta V_{\varphi, \pi}^i(s) \propto\ \mathbb{E}\_{\varphi,\pi} \left[ W\_{\varphi,\pi}^i(s, a) \cdot \left[ \nabla_\eta \log \pi_{\theta}(a \mid o, \sigma) + \nabla_\eta \log \varphi_\eta(\sigma \mid s, o) \right] \right].$$
>
> And if using the vanilla policy gradient, it would be:
> $$
> \mathbb{E}\_{\varphi,\pi} \left[ Q\_{\varphi,\pi}^i(s,\sigma) \cdot \nabla_\eta \log \varphi_\eta(\sigma \mid s) \right].
> $$
> One could see that vanilla policy gradient is biased by missing the $\nabla_\eta \log \pi_{\theta}(a \mid o, \sigma)$ term.
>
> ### Q6: Why $\nabla_\eta \pi_{\theta}(a\mid o,\sigma) =0$ should be 0?
>
> We assume that the reviewer is referring to line 235 of the manuscript. In obedience constraints, $\sigma$ and $\sigma'$ are given by the forall quantifier, see Equation (5). Therefore $\sigma$ and $\sigma'$ are not given by $\varphi_\eta$. As such the gradient does not backpropagate to $\eta$.
>
> ### Q7: Why should the sender's signaling scheme and the receiver's action policy be updated alternately rather than simultaneously?
>
> The point of being updated alternately is indeed not explicitly mentioned in the paper. Thank you for pointing this out, and we will include this clarification in the manuscript.
>
> This is achieved through the use of **online cross-validation** technique, which is used to optimize hyperparameters in meta-gradient RL, and by the optimal reward framework. In fact, this technique is inspired by LIO (Yang, Jiachen, et al. "Learning to incentivize other learning agents." *NeruIPS (2020).). We are somewhat aligned with LIO, and the discussion of these perspectives is presented in Section 4.5.4: Hyper Gradient.
>
> Additionally, the earliest instance of online cross-validation can be traced back to: Sutton, Richard S. "Adapting bias by gradient descent: An incremental version of delta-bar-delta." *AAAI*. 1992.
>
> ### Q8: What are the feasibility, complexity, and difficulty of building a reputation system to support the basic assumptions of the new algorithm and the Markov signaling game?
>
> This refers to some future directions and is out of the scope of this manuscript.

---

> ### Comment · Reviewer_YNGH · 2023-08-11
> **Thank the authors for their response**
>
> I would like to thank the authors for providing response to my concerns. I will raise my recommendation a bit. While the response addressed some of my concerns, I don't fully agree with the claim that, during centralized training, the receiver can genuinely provide its internal information to the sender. If the receiver and the sender have potential conflict of interest, we cannot train them centrally since they belong to different parties. The discussion regarding why the signalling gradient can alleviate the non-stationarity issue does not seem to provide further insights on the new algorithm design and its novelty.

---

> > ### Author Response · Authors · 2023-08-12
> > **Thank the reviewer YNGH for their prompt reply and the change in rating**
> >
> > Thank you for your prompt reply and the change in rating encourages us very much. We wanted to further explain our understanding of the access of information.
> >
> > The main objective of the algorithm is learning to persuade others. The sender and receiver play different roles in training. Sender's signaling scheme is the main objective to train, while the receiver is a "dummy" learning agent that is auxiliary in centralized training.
> >
> > Once a sender agent is trained, it is no longer connected to the receiver. One could deploy such a sender agent to real use cases. For example, one could train an information design algorithm for vehicle routing using simulated vehicles in training, and deploy it for real vehicles. As long as the signaling scheme is persuasive, it is supposed to work for rational, self-interested agents.
> >
> > Therefore, our algorithm does not require access to those real agents' (e.g. real vehicles') policies. Instead, our algorithm only requires assumptions that they are rational and self-interested (which are commonly used in computational economics) and simulate them as such.
> >
> > In fact, this insight is related to the commitment assumption, which is removed in our work. This assumption suggests that after the sender identifies a suitable signaling scheme, it can commit this scheme to the receiver before any communication takes place. The receiver can then calculate that this signaling scheme is advantageous to itself, leading it to trust the sender. The signaling scheme derived using information design methods satisfies incentive compatibility. In other words, it ensures that the receiver has posterior trust in the sender's communicated information (achieved through obedience constraints), which guarantees the receiver's trust.
> >
> > In summary, here is the full process of training and deploying a signaling scheme:
> >
> > 1. Through centralized training, the sender and the simulated receiver establish an effective signaling scheme to gain the simulated receiver's trust.
> > 2. The sender then commits this signaling scheme to the actual interacting receiver.
> > 3. The actual receiver, based on calculations, determines that the signaling scheme promised by the sender is trustworthy. As a result, the actual receiver decides to adopt the action policy trained using this signaling scheme.
> >
> > Since both the signaling scheme and the action policy are consistent with the training, the expected performance and outcomes align with the trained results. At this point, centralized training only occurs between the sender and the simulated receiver, rather than the actual receiver.
> >
> > Two additional pieces of remarks:
> >
> > - In situations with mixed motives, there are equilibria that benefit both parties rather than being completely adversarial. Some equilibria lead to mutual gains and thus the receiver has the potential intension to offer its information.
> > - Although MADDPG involves competitive tasks, it can still justify that competitive and mixed-motive tasks can be addressed through centralized training and decentralized execution.

---

> > > ### Comment · Reviewer_YNGH · 2023-08-18
> > > **Thank the authors for further clarifications**
> > >
> > > Thank the authors for the further clarifications. I was wondering whether the scenario of a trained sender and a potentially adversarial and separately trained receiver has been extensively studied in the paper? It would be great if the authors can clarify the relevant experiment results reported in the paper.

---

> > > > ### Author Response · Authors · 2023-08-19
> > > > **Response**
> > > >
> > > > We thank the reviewer for the further question.
> > > >
> > > > We conducted a new experiment that does this training in two phases. The environment is Recommendation Letter.
> > > >
> > > > Phase 1: 1 sender and 1 receiver are trained.
> > > >
> > > > Phase 2: The sender trained in Phase 1 is fixed. 1 learning receiver (initialized randomly) is trained by interacting with this (fixed) sender through policy gradient.
> > > >
> > > > We observe that Phase 1 is similar to the experiments in the manuscript. In Phase 2, the new, unseen receiver quickly converges to the behavior of the receiver in Phase 1. This means that the (fixed) sender could persuade a new, unseen receiver through a period of interaction. We find the results consistent with our expectations.
> > > >
> > > > As per the rebuttal rules we are unable to update the rebuttal pdf to include more figures. We cannot provide anonymous links in the rebuttal either. However we have sent the anonymous links of the figures of this experiment to the area chair, who may share the link in to the reviewers if deemed appropriate. Curves of the same color in the graphs represent results from runs with the same seed. 10 random seeds are tested.
> > > >
> > > > We are happy to have further discussions with the reviewers.

---

### Official Review · Reviewer_6N28 · 2023-07-02

**Soundness:** 2 fair
**Presentation:** 2 fair
**Contribution:** 2 fair
**Rating:** 5
**Confidence:** 3

**Summary:**

This paper proposes a new agent-interaction framework called Markov signaling games, and the corresponding algorithm for mixed-motive tasks. Various experiments and discussions are provided to show the efficacy of the proposed method.

**Strengths:**

- The authors provide a new formulation of agent-interaction framework for multi-agent reinforcement learning and the corresponding detailed explanation.

- The proposed algorithm to solve Markov signaling game with theoretical results is provided.





**Weaknesses:**

It is unclear what makes Markov signaling games important. While related works on multi-agent RL with communication are provided, It seems that Markov signaling games appear to be just one concept of communication in multi-agent RL. In the proposed game, it is assumed that the sender observes a fully environmental state and sends a message to the receiver. Is this assumption reasonable and important? The authors should explain why Markov signaling games are important compared to conventional communication frameworks in multi-agent RL, where multiple agents aim to solve tasks while communicating with other agents. The conventional framework does not even differentiate between which agents are senders or receivers.

Furthermore, the paper should include a comparison with state-of-the-art multi-agent communication algorithms such as TarMAC[1], ATOC[2], and others. Currently, the authors only provide a comparison with DIAL.


[1] A. Das et al., "TarMAC: Targeted multi-agent communication"

[2] J. Jiang et al., "Learning attentional communication for multi-agent cooperation"

**Questions:**

See Weakness

**Limitations:**

The limitations are provided in Appendix.

---

> ### Author Rebuttal · Authors · 2023-08-10
>
> We thank the reviewer for their insightful review. We now provide our responses. We wanted to point out that our work is a learning-based extension of information design in sequential decision-making. It focuses on mixed-motive settings which are less addressed in existing MARL communication works. We also wish that with our responses and explanation, the reviewer might re-evaluate the work should they have their concerns addressed.
>
> ### What makes Markov signaling games important?
>
> MSG is the necessary and fundamental formulation to achieve information design. Please kindly refer to the global response for a detailed explanation.
>
> ### Comparison with SOTA MARL communication algorithms
>
> We believe that additional experiments are not necessary because it is not relevant to the scope of our work. We are not aiming to improve algorithm performance based on existing research on cooperative settings. Instead, we focus on mixed-motive settings.
>
> First, let's take the Recommendation Letter experiment (detailed description can be found in Section 5.1) as an example. This task is already quite simple, and DIAL has reached the limit of traditional MARL communication methods. In fact, theoretically designed algorithms for fully cooperative tasks cannot be applied to mixed motive tasks and cannot reach equilibrium in theoretical analysis. These algorithms are not fitting into mixed-motive tasks, and the training for these tasks is not even well-defined.
>
> For instance, the insight behind DIAL is to update the sender's signaling net based on the receiver's gradient, but this means that the training only utilizes information from $r^j$ and does not leverage $r^i$. As a result, the performance of $r^i$ in the experiments is significantly poor. This issue does not arise in fully cooperative tasks because all agents have the same reward.
>
> In other AC-framework-based algorithms, they use independent policy gradient, which is biased compared to our signaling gradient, and we have provided experiments comparing PG with SG. If we were to compare other AC-framework-based algorithms, we could simply replace their PG part with SG and add the Lagrange term for obedience constraints. **Conducting experiments to compare with DIAL's performance was already done for completeness**, and there is no need to dwell on supplementing experiments for fully cooperative tasks.
>
> Additionally, the notion of "mixed-motive" that we are referring to is different from that in TarMAC and ATOC. In their mixed motive task, predator and prey teams are fully cooperative within each team but fully competitive between teams. However, their algorithms focus on within-team interactions, so they essentially concentrate on fully cooperative scenarios. On the other hand, the problem we are discussing involves individual agents with mixed motives, where **mixed motives exist between the two agents themselves**.
>
> ### Added explanation and comparison with social influence-based MARL communication algorithms
>
> We thank the reviewer for suggesting exploring more into the subarea of MARL communication. Indeed, there is an existing social influence-based work that conducts insightful and pioneer attempts on communication in mixed-motive tasks. Please kindly refer to the global response for a detailed description and comparisons.

---

> ### Author Response · Authors · 2023-08-14
> **Follow-up discussions**
>
> We thank the reviewer again for their effort in reviewing our manuscript. With the reviewer's help, we have revised our literature review on MARL communication. We have explained in the rebuttal that we believe *TarMAC* and *Learning attentional communication* are less relevant in mixed-motive settings, while we have added discussions and experiments on social influenced-based communication methods for mixed-motive tasks. We wanted to learn if the reviewer has further questions and comments.
>
> We very much enjoy the discussions with the reviewers, and look forward to having your reply.

---

> > ### Comment · Reviewer_6N28 · 2023-08-17
> > **Response**
> >
> > Thank the authors for the detailed response. Most concerns are addressed and thus I raised the score.

---

> > > ### Author Response · Authors · 2023-08-18
> > > **Thank you**
> > >
> > > We thank the reviewer for sharing the concerns and having the discussions. We are glad to hear that most concerns are addressed and the raised score encourages us very much.

---

### Official Review · Reviewer_5nfz · 2023-07-04

**Soundness:** 3 good
**Presentation:** 4 excellent
**Contribution:** 3 good
**Rating:** 6
**Confidence:** 3

**Summary:**

 This paper addresses the problem of multi-agent reinforcement learning in environments with mixed motives, which lie on the spectrum from fully cooperative to fully competitive scenarios. Specifically, it studies the problem of information design in this setting: how a sender might craft messages so that the behaviour of a receiver is successfully influenced. There are non-trivial implications in this setup, including the non-stationarity introduced by the messages and the fact that the receiver may choose not to obey them. The authors propose an algorithm to derive a signaling policy that takes into account the effect of the messages on the receiver's policy, as well as a way to learn the receiver's policy in a way that provides compatibility between the incentives. The algorithm is validated on two types of environments with mixed incentives.

**Strengths:**

Originality: the paper is original in its consideration of mixed-motive environments, which have not received much attention in MARL, and yet are highly relevant for practical scenarios.

Quality: the paper is of good quality. It motivates the work well by drawing connections to computational economics. It proposes methods that are grounded and sound, and performs a reasonable evaluation.

Clarity: the paper is well-structured and clear overall, although I was not able to understand some of the technical details as I am not familiar with the area in depth.

Significance: the area is one of active interest in the NeurIPS community.

**Weaknesses:**

W1. In my opinion, the primary weakness of this work is the strength of a particular assumption made by the authors: that the sender has full access to the receiver's policy and observation at every step. I am struggling to think of a scenario in which the assumption is suitable (since it amounts to global visibility) and yet the setting is not fully cooperative. What are some examples of scenarios in which this is the case? This aspect needs to be discussed more thoroughly. Otherwise, the proposed method may be of limited practical use.

**Questions:**

QS1. Please see W1 above, I would appreciate an answer.

QS2. In the evaluation, it is unclear to me what is the appropriate metric to judge the performance of the algorithm. Is it $r^i$, $r^j$, or $r^i + r^j$ (social welfare)? Can we even speak of a right metric given the incompatibility between the incentives? Given that e.g. DIAL reaches the best equilibrium for the receiver, are the methods even comparable, or is the evaluation merely meant to illustrate which equilibria are reached? The "goal" of the evaluation should be clarified.

QS3. Note that the supplementary material also includes the main paper text, and that the main PDF is not rendered properly (text is inaccessible, references are not linked). Consider compiling the main paper and supplementary material separately instead of cutting the PDF.

QS4. Some nitpicks: first phrase of the abstract, while RL definitely is inspired by reward-driven learning in humans and animals, it is not necessarily setting out to mimic it; line 90 nature -> environment?.


**Limitations:**

Code and detailed information is available, which means the work is reproducible. No limitations beyond those already discussed in previous points. The authors have discussed broader impacts. The supplementary material includes a high-quality discussion of possible extensions in great depth.

---

> ### Author Rebuttal · Authors · 2023-08-10
>
> We thank the reviewer for their insightful review. We now provide our responses.
>
> ### QS1: Is it reasonable for the sender to access the receiver's actor and observation during training? What is the practical use of this model?
>
> Yes, it is reasonable for the sender to access the receiver's actor and observation during training. Please kindly refer to the global response.
>
> To illustrate its applications, we cited examples from this survey: Kamenica, Emir. "Bayesian persuasion and information design." *Annual Review of Economics* (2019).
>
> "Research in this second strand includes applications to (in no particular order): financial sector stress tests, **grading in schools**, employee feedback, law enforcement deployment, censorship, entertainment, financial over-the-counter markets, **voter coalition formation**, research procurement, contests, medical testing, medical research, matching platforms, price discrimination, financing, insurance, transparency in organizations, and **routing software**."
>
> In fact, the classic Recommendation Letter (which serves as the experimental environment to test our algorithm. Detailed explanation can be found in **Section 5.1**.) serves as a real-world example corresponding to the "grading in schools" mentioned earlier.
>
> Furthermore, **routing software** is another easily imaginable example. In fact, this is the work we are currently undertaking (and this paper serves as its theoretical foundation). The central routing software and user vehicles are in a **mixed-motive** scenario, where the routing software aims to optimize the overall traffic speed, while each user wants to increase their individual speed. Moreover, the routing software has more **informational advantage** than user vehicles; it can know the congestion status of all relevant road segments, while user vehicles are not allowed to do so. Each user is self-interested, which may lead to a decrease in collective benefits, known as a social dilemma. A common example of this is **Braess's paradox**. Using information design can resolve this issue. For specific details, please refer to the article mentioned in this paper: Das, Sanmay, Emir Kamenica, and Renee Mirka. "Reducing congestion through information design." *2017 55th annual allerton conference on communication, control, and computing (allerton)*. IEEE, 2017.
>
> In recent years, there has been significant attention on autonomous driving, but fully cooperative communication algorithms cannot meet the requirements of this scenario. We believe that information design is a crucial approach to address the communication challenges in this context, making our paper highly valuable.
>
> ### QS2: What is the appropriate metric to judge the performance of the algorithm?
>
> We thank the reviewer for this insightful question. We have the below response and will add the discussions to the main body of the paper.
>
> First, our designed algorithm takes the perspective of a self-interested sender, aiming to optimize the following problem:
> $$
> \begin{aligned}
> \max\limits_{\varphi} \mathbb{E}_{\varphi}[\ r^i(s, a) \ ],\quad\textrm{s.t. Obedience Constraints.}
> \end{aligned}
> $$
> Therefore, the measure of success for our algorithm primarily depends on $r^i$. However, we also present the curve of $r^i+r^j$ (social welfare) in an attempt to demonstrate that **even when the sender is self-interested and the signaling scheme is deceptive, it does not harm social welfare.**
>
> Moreover, for the same experiment, if we add a constant to all sender rewards, the sender can still persuade the receiver using the same signaling scheme (as the receiver's reward remains unchanged), and thus the receiver's utility remains unaffected. In this case, the difference between fully cooperative communication methods (e.g., DIAL) and our method becomes evident. Because DIAL only considers the receiver's reward and does not utilize the sender's reward (according to their concept, their method does not even use $r^i$ and does not reflect the self-interested sender). **If we add a constant to all sender rewards, their method's social welfare remains unchanged, while our method's social welfare improves.**
>
> Also, the specific experimental results show that, even in the simplest scenario like Recommendation Letter, where the sender is self-interested, our method's social welfare far surpasses that of DIAL.
>
> For completeness, we discuss the case where the sender's optimization objective is social welfare in lines 134-137 and in Appendix I: Broader Impacts. In this case, although the sender receives rewards from the environment ($r^i$), the sender can consider the following problem:
> $$
> \begin{aligned}
> \max\limits_{\varphi} \mathbb{E}_{\varphi}[\ r^i(s, a)+r^j(s, a) \ ],\quad\textrm{s.t. Obedience Constraints.}
> \end{aligned}
> $$
> And in this setting, comparing social welfare is fair.
>
> Most of the existing MARL communication methods are not for mixed-motive tasks. The insight of DIAL is to update the sender's signaling network based on the receiver's gradient, but this means that the training only utilizes information from $r^j$ and does not take advantage of $r^i$. In fully cooperative tasks, this is not an issue because all agents' rewards are always the same.
>
> We thank the reviewer for QS3 and QS4 and we have fixed them immediately.

---

> > ### Comment · Reviewer_5nfz · 2023-08-15
> > **Post-rebuttal response to authors**
> >
> > Many thanks, I appreciate the very detailed and precise response. The Braess paradox example is a great one and, in my opinion, more intuitive than the recommendation letter example. Possibly adding it in the introduction can help clarify the setting. Re. QS2, I'd also suggest adding this detail to the manuscript.
> >
> > Given the current limited scalability of the method, I am sticking with the original score. However, the treatment of the problem is original and I would encourage the authors to pursue the information design problem in a larger-scale MARL setting.

---

> > > ### Author Response · Authors · 2023-08-15
> > > **Thank you for the response**
> > >
> > > We thank the reviewer for the response and again for their feedback. All suggestions are well taken and we have updated the manuscript.
> > >
> > > We are also glad to hear that the reviewer decides to maintain their positive evaluation. This encourages us very much. We do plan to further pursue mechanism and information design methods into large-scale MARL problems and real applications.

---

### Official Review · Reviewer_bi5b · 2023-07-04

**Soundness:** 3 good
**Presentation:** 3 good
**Contribution:** 2 fair
**Rating:** 6
**Confidence:** 1

**Summary:**

The article tackles the scenario where two agents, a sender $i$ and a receiver $j$, act in an environment.
The sender $i$ can send a message to the receiver $j$ at each time step.
The sender's goal is to learn what information to send to the receiver such that it influences the latter's policy to maximize its own return.
The authors derive a learning signal analogous to the policy gradient in the single-agent setup, and they also propose a version with a supplementary obedience constraint.



**Strengths:**

The strength of the paper lies in the extensive discussion of the setup and its challenges, and the different equilibria that could be reached by the two agents. These are also supported by empirical results.

**Weaknesses:**

The scenario is limited to the case where the sender can backpropagate through the receiver's estimator, which looks far from a realistic setup.

Formula (3) needs some introduction. It is not clear why that formula expresses an obedience constraint. Similarly, it would be helpful to restate the revelation principle, not just mention it.

**Questions:**

What is incentive compatibility?

**Limitations:**

The authors mention some limitations, but I would add to that the need to backpropagate through the receiver's policy estimator.

---

> ### Author Rebuttal · Authors · 2023-08-10
>
> We thank the reviewer for their insightful review. We now provide our responses.
>
> The suggestions by the reviewer have reminded us that we should provide more detailed explanations for some fundamental concepts, which could also be confusing for other reviewers. We have improved the explanations of these concepts in the main text.
>
> ### Is it reasonable for the sender to access the receiver's actor during training?
>
> Yes, it is reasonable. Please kindly refer to the global response.
>
> ### Introducing obedience constraints and the revelation principle.
>
> In the context of information design, the revelation principle can be understood as follows: **The revelation principle states that there exists an optimal signaling scheme that does not require more signals than the number of actions available to the receiver.** (As mentioned in line 200 of our paper.) This principle is commonly used in the field of information design.
>
> The **obedience constraints** are the core of information design and it significantly improves the performance in experiments. Assuming that the sender's signal set is equal to the receiver's action set, the sender's signals can be interpreted as recommending the receiver to take a specific action (This common assumption is without loss of generality according to the revelation principle). Under this premise, **obedience constraints ensure that the receiver will definitely follow the sender's recommendations.**
>
> To explain why is it that as long as the sender's signaling scheme satisfies obedience constraints, the receiver will definitely follow the sender's recommendations, we provide a simple derivation as follows:
> $$
> \begin{aligned}
>   & \sum\limits_{s} \mu_0(s)
>   \cdot \varphi( a\mid s )
>   \cdot \Big( r^j(s, a) - r^j(s, a') \Big) \ge 0 \\
>   \Leftrightarrow &
>   \sum\limits_{s} \frac{\mu_0(s) \cdot \varphi( a\mid s )}
>   { \sum\limits_{s'}\mu_0(s') \cdot \varphi( a\mid s')}
>   \cdot \Big( r^j(s, a) - r^j(s, a') \Big) \ge 0 , \forall a'\in A.\\
>   \Leftrightarrow &
>   \sum\limits_{s} \mu(s\mid a)
>   \cdot \Big( r^j(s, a) - r^j(s, a') \Big) \ge 0 , \forall a'\in A.\\
>   \Leftrightarrow &
>   \sum\limits_{s} \mu(s\mid a)
>   \cdot r^j(s, a)  \ge
>   \sum\limits_{s} \mu(s\mid a)
>   \cdot r^j(s, a'), \forall a'\in A.
> \end{aligned}
> $$
> In the derivation, $\mu_0$ represents the prior probability, and $\mu$ represents the posterior probability. Therefore, a self-interested and rational receiver will definitely follow the sender's recommendations, because the posterior expected payoff of the action recommended by the sender is greater than or equal to the posterior expected payoffs of all other actions.
>
> This greatly simplifies the problem, allowing **the sender to choose the action that maximizes its expected payoff, while ensuring that the receiver obeys, and then recommends the receiver to take that action.** Thus, the specific representation of the sender's optimization goal is:
> $$
> \begin{aligned}
> \max\limits_{\varphi} \mathbb{E}_{\varphi}[\ r^i(s, a) \ ],\quad\textrm{s.t. Obedience Constraints.}
> \end{aligned}
> $$
>
>
> ### Introducing incentive compatibility
>
> Our setting considers convincing a **self-interested rational** receiver to take actions that benefit the sender in mixed-motive tasks.
>
> **Incentive compatibility** refers to the idea that based on this receiver's assumption (self-interested and rational), the sender needs to make the receiver **"believe"** that the recommended action provides the maximum payoff (in terms of **posterior** belief), so that the receiver will choose the action recommended by the sender.

---

### Official Review · Reviewer_sBPo · 2023-07-07

**Soundness:** 3 good
**Presentation:** 3 good
**Contribution:** 2 fair
**Rating:** 6
**Confidence:** 3

**Summary:**

This paper studies how multi-agent reinforcement learning agents influence each other by hsaring different kinds of information. Additionally, the authors study how different information affects the transition of the agents trajectories and what information can be ignored. To ensure efficient learning, a suitable gradient for the problem is also proposed.

**Strengths:**

This paper is well written and reads well. It is also well structured and contains only few spelling mistakes.
Some other strengths in particular:
- Proposes markov signalling games
- Provides an extensive theoretical analysis of the approach
- Develop the notions of a signalling gradient

**Weaknesses:**

1. some of the related works in the appendix could be moved (or summarised) to the main text; the provided related works in the main text are too scarce and do not provide enough relevant information that relates to the paper
2. the introduced markov signalling games seem to have some links with the famous Lewis games and other similar referencial games. Would be interesting to see a discussion around these concepts, as I cannot find it in the paper. Some related works also use reinforcement learning to learn communication involving similar games and concepts, such as [1], [2], [3], [4].

minor:
- line 323: "ours" -> "our"

Overall, this paper is interesting and well structured. Please find here and below some points that I would like the authors to comment on.

[1] https://openreview.net/pdf?id=O5arhQvBdH

[2] https://arxiv.org/pdf/1804.02341.pdf

[3] https://arxiv.org/pdf/1804.03980.pdf

[4] https://arxiv.org/pdf/1705.11192.pdf

**Questions:**

1. in line 306 it is stated that the receivers action policy is implemented using A2C. Is this the case for the sender too? Or do they share the same network as in multiple MARL approaches? This should be clarified in the paper.

**Limitations:**

1. It would be interesting to analyse the explainability of the messages exchanged by the receiver and speaker in these experiments, using the proposed method, as I can't find this is the paper and it is something very relevant when it comes to communication-driven games.

---

> ### Author Rebuttal · Authors · 2023-08-10
>
> We thank the reviewer for their insightful review. We now provide our responses.
>
> ### Emergent communication
>
> This reviewer shows particular interest in understanding how information is conveyed semantically, especially in the context of **emergent communication** and the emergence of language in multi-agent learning methods.
>
> We find this aspect quite intriguing, and our experimental results have indeed observed a relevant phenomenon. Actually, we have discussions in **Section 5.3.1** of the paper, although we were not aware of it as a research direction beforehand. It's essential to note that this is not our primary focus, but we acknowledge that we lack an introduction to this direction in the related work. We will **add a separate subsection to the related work** to address this.
>
> ### Comparisons of Markov signaling games with referential games.
>
> We were not aware of the concept of referential games beforehand. After understanding them, we believe that these games **have significant differences** compared to the Markov signaling games we proposed.
>
> Lewis signaling games and other similar referential games are usually **cooperative** tasks. The core of these tasks is how the sender efficiently conveys its information to the receiver, enabling the receiver to take **a specific action desired by both the sender and receiver.** The crucial aspect lies in how the sender conveys information to enable the receiver to understand the intended semantics. Therefore, **the sender aims to transmit as much of its available information as possible.** They train communication as a referential tool, and messages are used to disambiguate between different possible referents.
>
> Information design focuses on **mixed-motive** communication tasks. The receiver's action will determine the payoffs for both parties, but there may not be a specific action that is desired by both. In fact, the **self-interested sender**'s goal is not to maximize the transmission of its known information, but to **leverage its information advantage to confuse the self-interested receiver, so as to maximize its own payoffs.** The sender aims to make the receiver believe in its statements in terms of posterior probabilities, convincing the receiver that following the sender's instructions is beneficial for the receiver itself. **the sender persuades the receiver to take actions that are advantageous to the sender** (In fact, senders typically learn deceptive strategies).
>
> Our experimental results also indicate that the sender and receiver can indeed establish a communication protocol through our method, allowing mutual understanding, even in situations where they have mixed motives (for a detailed discussion, refer to the answer to L1 below).
>
> ### On the references provided by the reviewer
>
> These papers focus on algorithms related to how to more efficiently convey information, the emergence of language in communication, and the relationship between learning algorithms and human language. Though they are not directly related to our work, these papers have inspired us in some ways, and we now have insights into the relationship between information design and emergent communication research. We have added [1-4] as references.
>
> The most relevant one is probably the third paper [3], as they mention a negotiation environment that is indeed a mixed-motive task. The points of **difference** between their setting and Markov signaling games (ours) are as follows:
>
> 1. They communicate multiple times and then choose an action. In contrast, we make a decision on the action after each communication round, and this selected action affects future states, making it a problem on a Markov model.
> 2. Their communication is bidirectional, while ours is unidirectional.
> 3. Their agent's information advantage over the other is its own utility function that remains hidden. In contrast, our sender's information advantage lies in its ability to observe the $s-o$ part that the receiver cannot see, and this portion is of interest to the receiver (as it determines the receiver's utility).
>
> The **connection** between their work and ours is: The communication protocol we study uses the **linguistic channel** they mentioned, as it also satisfies two key properties:
>
> 1. Non-bindingness: Messages sent via this channel do not commit the sender to any course of action (nor state), because the sender has an empty action space. The communication is purely a cheap talk.
> 2. Unverifiability: There is no inherent link between the linguistic utterance and the true situation of the current state, meaning that the sender could potentially lie.
>
> With their approach, self-interested agents cannot achieve good results using a linguistic channel, while ours can achieve good performance in Markov siganling games.
>
> The fourth paper [4] reminded us that we also used **Gumbel-Softmax** to allow for end-to-end differentiation. Specifically, this is manifested in the signaling gradient derivation, where an unbiased gradient term includes $\nabla_\eta \log \pi_\theta(a\mid o,\sigma)$, and this gradient is passed from the receiver's action policy to the sender's signaling network. The missing reference is "Categorical Reparameterization with Gumbel-Softmax."
>
> ### Implemented using A2C and explainability of the messages
>
> We are using A2C **without parameter sharing**. In the experiments, PG represents the A2C algorithm; PGOC represents the A2C algorithm with extended obedience constraints. **SG represents the A2C algorithm with the policy gradient update part replaced by signaling gradient.** SGOC represents the SG algorithm considering extended obedience constraints when updating the signaling network.
>
> We did not focus on the explainability of the messages. As we have discussed in **Section 5.3.1**, in some training seeds, the sender uses the signal **'1'** to represent a good student, while in other seeds, the sender uses the signal **'0'**. Therefore the messages are not explainable.

---

> > ### Comment · Reviewer_sBPo · 2023-08-14
> > **Thanks for the response**
> >
> > I thank the authors for their detailed response. As the authors have stated, I think that it is important to include these discussions and references around emergent communication and signalling games, since they are strongly related to what is being presented. After these discussions and assuming that this is now included, I have no further questions and will raise my score.

---

> > > ### Author Response · Authors · 2023-08-14
> > > **Thanks for the reply and the re-evaluation**
> > >
> > > We thank the reviewer for their effort in replying to us and raising the score. It encourages us very much. We are glad to have our manuscript improved by including the references and discussions on emergent communication and signalling games.
> > >
> > > If the reviewer finds any further questions, we are more than happy to provide responses.

---

### Author Rebuttal · Authors · 2023-08-10

We notice some common concerns expressed by the reviewers. One is on the sender's access to the receiver, and the other is on MSG for mixed-motive MARL communication. These "concerns" are not issues and we are happy to explain why.

### Sender and receiver are in interest conflict. Can the sender have access to the receiver's policy and observation in signaling gradient (2)?

Definitely yes for accessing the policy. Such access is only used in centralized training (CT) and once it is trained the access is no longer needed in decentralized execution (DE). Therefore, when interest conflict applies (execution time), the access is not used.

In MARL algorithms, whether they are communication algorithms or not, the CTDE (Centralized Training with Decentralized Execution) framework is commonly used (see, e.g., **MADDPG**). This means that in the simulated environment, data is centralized and accessible during training, allowing us to use all the required quantities. As long as the algorithm does not require cross-agent access during agent execution, it is reasonable.

Additionally, in communication methods, it is common for gradients to pass through other agents during training. We adopted the commonly used **Gumbel-Softmax** technique in the field of emergent communication research to allow for end-to-end differentiation, which is used to retain the gradients of sampled signals. Examples are

For observation, the receiver's observation is **common knowledge** between the sender and receiver. This is quite reasonable, as the sender possesses an informational advantage. For example, in the case of vehicle routing, the routing algorithm will have access to the user vehicles' observations. We have discussed the case if access to the observation is not available in **Appendix B**: Extensions of Markov Signaling Games.

[1] Havrylov, Serhii, and Ivan Titov. "Emergence of language with multi-agent games: Learning to communicate with sequences of symbols." *Advances in Neural Information Processing Systems (2017).

[2] Foerster, Jakob, Ioannis Alexandros Assael, Nando De Freitas, and Shimon Whiteson. "Learning to communicate with deep multi-agent reinforcement learning." Advances in Neural Information Processing Systems (2016).


### Justification for our proposed process: Markov signaling games

The core of MSG is extending information design to sequential decision-making scenarios. Specifically, what makes MSG different from other communication processes can be summarized in the following points:

1. The sender and receiver are heterogeneous, with the sender having an **informational advantage**, which lies in its ability to observe the $s-o$ part that the receiver cannot observe, and this part is of interest to the receiver (as it influences the receiver's utility);
2. Their interests may differ. Their rewards are not specified to be identical, so they may be in a mixed-motive scenario. In reality, this is quite common, but the current MARL communication framework typically focuses on fully cooperative scenarios;
3. The sender cannot take actions, and its signal is not an influencing factor in anyone's reward function. Therefore, the sender can only influence the receiver's belief and subsequently modify its own expected payoffs by changing the signaling strategy;
4. The receiver can take actions to directly determine the payoffs for both parties, but lacks sufficient information to estimate the current state and thus needs to obtain information from the sender.

These differences are necessary to re-inspect the MARL communication process in a mixed-motive environment, and provide the necessary formulation to apply game-theoretic methods for the communication algorithm.

### Added discussions and experiments on existing mixed-motive communication works

While most of the works in communication do not directly extend to mixed-motive cases, Section 5 in [3] conducts insightful pioneer works on this setting. We apologize for missing the references in our original text.

In [3], the authors propose social influence, which is defined using KL divergence to identify actions that cause the greatest change in the action distribution of others (in other words, actions that have the most influence on others). The social influence is then treated as an intrinsic reward. In this way, the agent will try to influence others as much as possible. This method serves as a promising initial attempt for this type of problem, achieving favorable results in several experiments. However, whether/why maximizing influence on others leads to good performance in mixed-motive communication tasks remains unclear.

Information design is a dedicated subarea in economics that studies mixed-motive communication problems. Our Markov signaling games and the SGOC algorithm are armed with formulations in information design, and can be regarded as a more methodological and game-theoretic follow-up of [3]. Our method is sound as the extended obedience constraints will induce incentive compatibility, which will be the foundation that the receiver will respect the message.

We have re-implemented [3] and found out that it does not work in the Recommendation Letter task. The learning curves are provided in the attached PDF. We find that the receiver is unlikely to respect the message over time, which agrees with the following description in [3]: "Because the listener agent is not compelled to listen to any given speaker, listeners selectively listen to a speaker only when it is beneficial, and influence cannot occur all the time."

[3] Jaques, Natasha, Angeliki Lazaridou, Edward Hughes, Caglar Gulcehre, Pedro Ortega, D. J. Strouse, Joel Z. Leibo, and Nando De Freitas. "Social influence as intrinsic motivation for multi-agent deep reinforcement learning." In International Conference on Machine Learning (2019).

---

### Decision · Program_Chairs · 2023-09-21

**Decision:**

Accept (poster)

**Comment:**

After a lively discussion and detailed and thorough rebuttal by the authors, several reviewers of this paper (sBPo, 6N28, and YNGH) have raised their initial scores, leading to a consensus among the reviewers that the paper should be accepted. The scores are now 6,6,6,5,5, with an average of 5.6, which is just above the acceptance threshold.

Positive aspects of the paper identified by the reviewers include its **quality** ("It motivates the work well by drawing connections to computational economics. It proposes methods that are grounded and sound, and performs a reasonable evaluation."), **clarity** (“This paper is well written and reads well.”), and the **importance** of the problem (“It is interesting and important to study the information sharing problem in a multi-agent context. The Markov signaling game presented in this paper appears to be of some practical values and deserves further investigation.”).

Below, I will summarize the main themes of the concerns raised by the reviewers as well as the authors' rebuttal to these concerns. Overall, given the authors' response thoroughly addresses the reviewers' concerns, and the reviewers themselves have raised their scores to a unanimous accept, I recommend the paper be accepted.

### Lack of citations of relevant prior work
Several reviewers complained that the authors did not include a thorough related work section, and particularly that they failed to cite important works on multi-agent (emergent) communication, referential games, etc. In fact, the authors originally claimed that they were the first to study multi-agent communication in mixed motive settings, failing to cite prior work that does address this problem (e.g. https://arxiv.org/abs/1810.08647). In the rebuttal and global response, the authors have corrected these issues, added explanations of how the proposed work differs from this prior work, and even included additional experiments comparing to this baseline. So in my opinion, these concerns have been adequately addressed.

### Practical applicability of backpropagating through the receiver's policy
Several reviewers initially complained that because the method relies on backpropagating through the receiver's policy in a mixed motive setting, it did not seem applicable to any realistic scenario. Quoting reviewers:
- bi5b: “The scenario is limited to the case where the sender can backpropagate through the receiver's estimator, which looks far from a realistic setup.”
- 5nfz: “In my opinion, the primary weakness of this work is the strength of a particular assumption made by the authors: that the sender has full access to the receiver's policy and observation at every step. I am struggling to think of a scenario in which the assumption is suitable (since it amounts to global visibility) and yet the setting is not fully cooperative”
- YNGH: “I cannot immediately relate the two toy problems studied experimentally in this paper with any practically important scenarios where an effective machine learning-based solution of the Markov signaling game is essential.”

The authors rebutted these points in the global response by saying that they are using the common MARL training paradigm of CTDE (centralized training, decentralized execution), and therefore it is reasonable for the receiver to share its policy weights during training, if not execution. They added in the response to reviewer YNGH that their method is applicable to a vast array of real-world scenarios, including grading in schools, employee feedback, law enforcement deployment, etc.

Initially, I did not find these arguments convincing, because I also did not see how the CTDE paradigm was relevant in these scenarios. I was skeptical that students submitting homework have the opportunity to backprop through their instructor's brain when completing homework assignments. I agreed with reviewer YNGH's comment: "when the sender and receiver have potentially conflicting interests, it is entirely unclear why the receiver would be motivated to honestly support the sender by providing its internal information to the sender (in a conflicting scenario, the sender and receiver may not have the motivation to participate in a centralized training process).”

However, the authors provided further clarification in a final comment, where they explained that: the “sender's signaling scheme is the main objective to train, while the receiver is a "dummy" learning agent that is auxiliary in centralized training. [...] Therefore, our algorithm does not require access to those real agents' (e.g. real vehicles') policies. Instead, our algorithm only requires assumptions that they are rational and self-interested (which are commonly used in computational economics) and simulate them as such." In this case, it does make sense to use a simulated, dummy receiver to train the sender, and I find that this concern is also addressed adequately.